# Learn to Merge: Meta-Learning for Adaptive Multi-Task Model Merging

**Jun Chen** [1 2]  **Qin Zhang** [† 1]  **Weizhi Zhang** [3]  **Xiao Luo** [4]  **Philip S. Yu** [3]  **Ziyue Qiao** [† 2]

## Abstract

Model merging in the pretrain-finetune paradigm has proven effective by combining multiple fine-tuned models into one with multi-task capabilities. However, existing methods rely on fixed or manually tuned merging coefficients, making the unified model sensitive to the initial merging strategy and suboptimal for downstream adaptation. Thus, this paper proposes an innovative model merging framework called MetaMerging, a novel meta-learning algorithm to adaptively optimize the merging coefficients to construct a unified model tailored for task-specific adapter training. By simulating adapter updates in an inner loop and meta-optimizing merging coefficients in an outer loop, MetaMerging produces more balanced and generalizable unified models. Extensive experiments on CV and NLP fields show strong performance of MetaMerging on various downstream tasks and demonstrate the effectiveness of meta-learning in our method compared to other parameter merging methods. Our code is available at https://github.com/cjcj46262/MetaMerging.

## 1. Introduction

With the rapid development of deep learning, model architectures are scaling up, and the computational burden of training is increasing (Krizhevsky et al., 2009; Dosovitskiy et al., 2020; Vaswani et al., 2017; He et al., 2016). Consequently, the pretrain-finetune paradigm has become a widely adopted approach, where a foundation model is first pretrained and then finetuned for multiple downstream

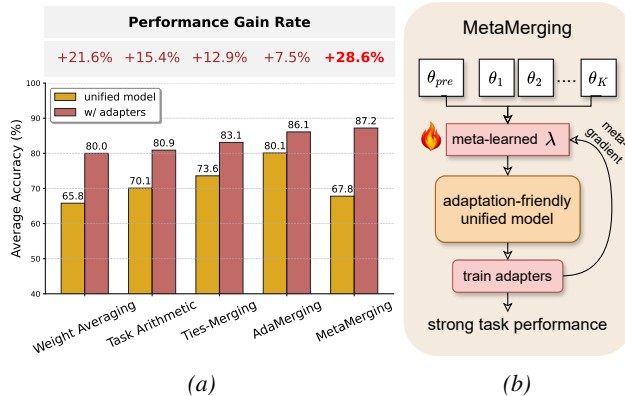

*Figure 1.* (a) Multi-task performance of unified models from different merging methods on eight vision tasks, evaluated before and after task-specific adapter training. (b) Diagram of our meta-learning process. The merging coefficients are optimized to produce a more generalized unified model, which enables more effective task-specific adapter training.

tasks (Liu et al., 2024; Wortsman et al., 2022b; Li et al., 2025; He et al., 2022). This usually results in multiple fine-tuned models, while maintaining separate models for multiple tasks results in substantial storage and computational costs. To address this, model merging has been proposed as an effective method for constructing a unified multi-task model by leveraging the knowledge of multiple finetuned models, and has quickly become a popular research direction (Yang et al., 2024a). This paradigm of "learning from models" (Zheng et al., 2025) is viewed as a strong complement to the traditional "learning from data," as merged models can effectively consolidate the knowledge inherent in finetuned models while enhancing overall parameter efficiency.

Early model merging methods (such as Weight Averaging (Wortsman et al., 2022a), Task Arithmetic (Ilharco et al., 2023), Ties-Merging (Yadav et al., 2023)) mainly focused on parameter-level operations, which are simple and efficient but often lead to a prominent performance gap between the merged model and the individually finetuned models. Such techniques are built on the task vector, which serves as a unique representation for a particular task and can be simply represented as $V_k = \theta_k - \theta_{pre}$, where $\theta_k, \theta_{pre}$ denote the parameters of finetuned model of task $k$ and pretrained model, respectively. The unified model is then computed by linearly combining these task vectors with merging

---

[†]Corresponding authors. [1]College of Computer Science and Software Engineering, Shenzhen University, China [2]School of Computing and Information Technology, Great Bay University, China [3]Department of Computer Science, University of Illinois at Chicago, USA [4]Department of Statistics, University of Wisconsin–Madison, USA. Correspondence to: Ziyue Qiao <zyqiao@gbu.edu.cn>, Qin Zhang <qinzhang@szu.edu.cn>.

*Proceedings of the 43ʳᵈ International Conference on Machine Learning*, Seoul, South Korea. PMLR 306, 2026. Copyright 2026 by the author(s).

coefficients $\{\lambda_k\}_{k=1}^K$, as Eq. 1. More recently, advanced merging methods (such as Surgery (Yang et al., 2024b), Twin-Merging (Lu et al.), WEMoE (Shen et al., 2024)) have introduced lightweight unsupervised training and additional components (e.g., task-specific weights, router modules) to enhance the merged model and improve performance. In general, these methods follow two sequential steps: ❶ merging the parameters of finetuned models into a unified model through conventional merging methods; ❷ training additional adapter modules based on the unified model. For example, in the Surgery method, the final multi-task model is constructed from two components: the unified model and task-specific adapters. This design separates shared knowledge from task-specific adaptations, improving scalability and parameter efficiency.

However, these methods emphasize the subsequent training phase but naively treat the merging coefficient $\{\lambda_k\}_{k=1}^K$ as manually tuned hyper-parameters or a uniform value $1/K$ (i.e., Weight Averaging). As shown in Figure 1a, under the same adaptation process, unified models produced by different merging strategies benefit unevenly from adapters, leading to substantial discrepancies in final performance. This is because the existing progressive knowledge modularization strategy faces significant limitations. First, the quality of the unified model heavily depends on the initial merging methods. If the merging process fails to properly balance the representations of each task, the unified model may favor certain tasks while neglecting others. In such cases, adding additional adapter modules may not fully compensate for the representation deficiencies of the unified model in certain tasks. Second, these methods lack the ability to consider the training needs of downstream modules during the merging process, leading to a disconnect between the merging results and task-specific adaptations, and failing to form effective synergy. Thus, a key research question arises: *How can we learn a strong initialization for the unified model, tailored to the training needs of downstream adapters, thereby achieving more balanced and effective multi-task performance*?

Motivated by these observations, we introduce a meta-learning algorithm and adaptively learn better merging coefficients to obtain a more generalizable and task-balanced unified model, which can better support the training of subsequent task-specific adapters for the given tasks, as illustrated in Figure 1b. The inspiration comes from MAML (Finn et al., 2017), a meta-learning algorithm that learns a good model initialization such that it can be quickly adapted to new tasks with only a few gradient updates. In this paper, we propose a novel model merging framework, MetaMerging, which constructs the merged model by integrating a unified model with task-specific adapters. In the inner loop, we perform rapid provisional gradient updates to imitate adapter training, and in the outer loop, we meta-update the

merging coefficients to improve inference performance by simulating test-time adapter usage, as shown in Figure 2. In this way, we can learn the best way to merge task vectors and automatically find the most generalized unified model, thereby enhancing the final performance after incorporating adapters. Interestingly, as the results of MetaMerging shown in Figure 1a, we found the optimal unified model is not necessarily the one that performs best on all tasks, but the one that better preserves the potential for adapters, thereby enabling greater downstream gains.

To summarize, our contributions are as follows: (1) We propose MetaMerging, an innovative model merging framework that substantially improves the multi-task performance of the merged model and is effective across a wide range of tasks. (2) We develop a meta-learning algorithm that autonomously learns to merge task vectors with proper coefficients, introducing a new direction for designing more effective and sophisticated model merging methods. (3) We conduct extensive experiments on various tasks and models, verifying the effectiveness of MetaMerging and enhancing its interpretability.

## 2. Related Work

**Model Merging** Recently, model merging has gained attention as a viable alternative to conventional multi-task and transfer learning paradigms across diverse scenarios, including computer vision, natural language processing, and graph learning (Yang et al., 2024a; Wan et al., 2024; Tang et al., 2024b; Xiong et al., 2024; Zhou et al., 2024a; Chen et al., 2026; Wang et al., 2025; Qiao et al., 2025). Instead of jointly training multiple tasks or relying on continual learning schemes, model merging directly constructs a unified model using the existing parameters of task-specific finetuned models. A straightforward strategy for model merging is to average the model weights (Wortsman et al., 2022a), but this often leads to substantial performance degradation. Early methods such as Task Arithmetic and Ties-Merging mainly operate in the parameter space, for example, by adjusting task-vector coefficients or applying sparsification, in order to alleviate task conflicts and improve the merged model's performance and generalization (Ilharco et al., 2023; Yadav et al., 2023; Yang et al., 2024d; Yu et al., 2024). Subsequent works have introduced more sophisticated merging strategies, such as task-specific module, optimal transport in parameter space, and router-based method, aiming to further narrow the performance gap between merged and finetuned models (Chen & Kwok, 2024; Lu et al.; Shen et al., 2024; Huang et al., 2024; Yang et al., 2024c). They typically follow the merging-and-training paradigm introduced in Section 1, which requires only the input data but not the labels of downstream tasks. Building upon such advanced structures, our method leverages meta-

learning to produce a more generalizable unified model, which effectively coordinates parameter merging with subsequent training to achieve superior performance.

**Meta-Learning** Meta-learning, also called "learning to learn", provides a general framework to acquire transferable knowledge across tasks (Han et al., 2021; Khoee et al., 2024; Vettoruzzo et al., 2024; Finn et al., 2018). The essence of meta-learning is that, rather than addressing tasks independently with a pre-defined algorithm, it improves the learning procedure by exploiting knowledge gained over repeated learning episodes (Hospedales et al., 2021). Classical approaches, such as meta-metric, meta-optimization, and model-based meta-learning, have been widely applied in few-shot and multi-task settings (Yuan et al., 2020). Among them, a classical method, MAML (Finn et al., 2017), aims to train initialization parameters that can be rapidly adapted to new tasks with limited data, while task-specific learners are guided by meta-gradients. Recent advances have extended meta-learning to parameter-efficient adaptation, neural architecture search, and multi-modal scenarios (Zhou et al., 2024b; Wang et al., 2022). In this paper, meta-learning has been used to learn the merging coefficients when combining task vectors, enabling a more adaptive and generalized unified model and more effective subsequent adapter training.

## 3. Methodology

### 3.1. Preliminaries and Motivation

We consider the standard pretrain–finetune paradigm with a pretrained backbone $f_{\theta_{pre}}$. For $K$ downstream tasks $\{1, ..., K\}$ with datasets $\{\mathcal{D}_1, \mathcal{D}_2, ..., \mathcal{D}_K\}$, we obtain $K$ task-specific finetuned models $\{f_{\theta_1}, ..., f_{\theta_K}\}$ from the same pretrained initialization $\theta_{pre}$. Following prior model merging work, we represent each task-specific finetuned model by its task vector $V_k = \theta_k - \theta_{pre}, k = 1, ..., K$ and construct the unified model as:

$$\theta_{uni}(\lambda) = \theta_{pre} + \sum_{k=1}^{K} \lambda_k V_k. \quad (1)$$

where $\lambda = \{\lambda_k\}_{k=1}^{K}$ are the merging coefficients controlling the balance across tasks. Note that task-specific prediction heads are not merged: the unified backbone $\theta_{uni}$ is shared across all tasks, while each downstream task retains its own head, forming a shared-backbone, multi-head architecture for multi-task inference.

Modern merging-and-training pipelines further attach lightweight task-specific adapters on the unified model and train them using only unlabeled data (Lu et al.; Shen et al., 2024; Yang et al., 2024b). Concretely, denote the adapter for task $k$ as $A_{\theta_{ada,k}}(\cdot)$. Given an input batch $\mathbf{X}$ sampled from task data, we align the representations of the adapted

unified model with the corresponding finetuned model:

$$\mathcal{L}_k(\mathbf{X}; \lambda, \theta_{ada,k}) = \left\| f_{\theta_k}(\mathbf{X}) - A_{\theta_{ada,k}}(f_{\theta_{uni}}(\mathbf{X})) \right\|_2^2. \quad (2)$$

This adapter training can be viewed as a lightweight test-time adaptation stage. The task-specific adapters serve as task-specific knowledge to narrow the "representation bias" comparing the finetuned models (Yang et al., 2024c). The merged model with adapters is expected to support multi-task inference on $K$ downstream tasks, which significantly reduces the total number of parameters, alleviating both storage and computational costs.

**Static Merging is Theoretically Misaligned with Post-Adaptation** A key design choice in model merging is the coefficients $\lambda$. A crucial observation is that $\theta_{uni}(\lambda)$ is *not* the final model that will be used for each task. Instead, it serves as an *initialization* that will be further adapted by training $A_{\theta_{ada,k}}$. Thus, selecting $\lambda$ by optimizing the performance of the unified model before adaptation is generally misaligned with the final performance after adaptation. We now formalize this mismatch by contrasting two objectives.

**Static Objective** If we evaluate the merged model directly, the coefficient learning problem is

$$F(\lambda) = \sum_{k=1}^{K} \mathcal{L}_k^{\text{task}}(\theta_{uni}(\lambda)). \quad (3)$$

where $\mathcal{L}_k^{\text{task}}$ denotes the task loss induced by the downstream task head.

**Post-Adaptation Objective** In contrast, the actual pipeline evaluates the model *after* task-specific adaptation. To capture the essence of parameter-efficient adaptation, we model downstream adaptation for each task $k$ as updates restricted to an affine subspace:

$$\theta \mapsto \theta + u, \quad u \in \mathcal{U}_k, \quad (4)$$

where $\mathcal{U}_k \subseteq \mathbb{R}^d$ is a proper linear subspace. Then we evaluate the merged model after training task-specific adapters:

$$G(\lambda) = \sum_{k=1}^{K} \min_{u \in \mathcal{U}_k} \mathcal{L}_k^{\text{task}}(\theta_{uni}(\lambda) + u). \quad (5)$$

**Theorem 3.1** (Static-Optimal Merging Is Generally Not Post-Adaptation Optimal). *Consider $K$ tasks and a unified model $\theta_{uni}(\lambda)$ defined in Eq. 1. For each task $k$, let $\theta_k^\star$ be a minimizer of a twice-differentiable loss $\mathcal{L}_k^{\text{task}}$. Assume that there exists at least one pair $(k, \lambda)$ such that $e_k(\lambda) = \theta_{uni}(\lambda) - \theta_k^\star$ has a non-zero component outside $\mathcal{U}_k$.*

*Then, for $\theta_{uni}(\lambda)$ sufficiently close to $\theta_k^\star$, the post-adaptation loss admits the local expansion*

$$\min_{u \in \mathcal{U}_k} \mathcal{L}_k^{\text{task}}(\theta_{uni}(\lambda) + u) = \mathcal{L}_k^{\text{task}}(\theta_k^\star)$$
$$+ \frac{1}{2} \left\| (I - P_k) e_k(\lambda) \right\|_{H_k}^2 + o(\|e_k(\lambda)\|^2). \quad (6)$$

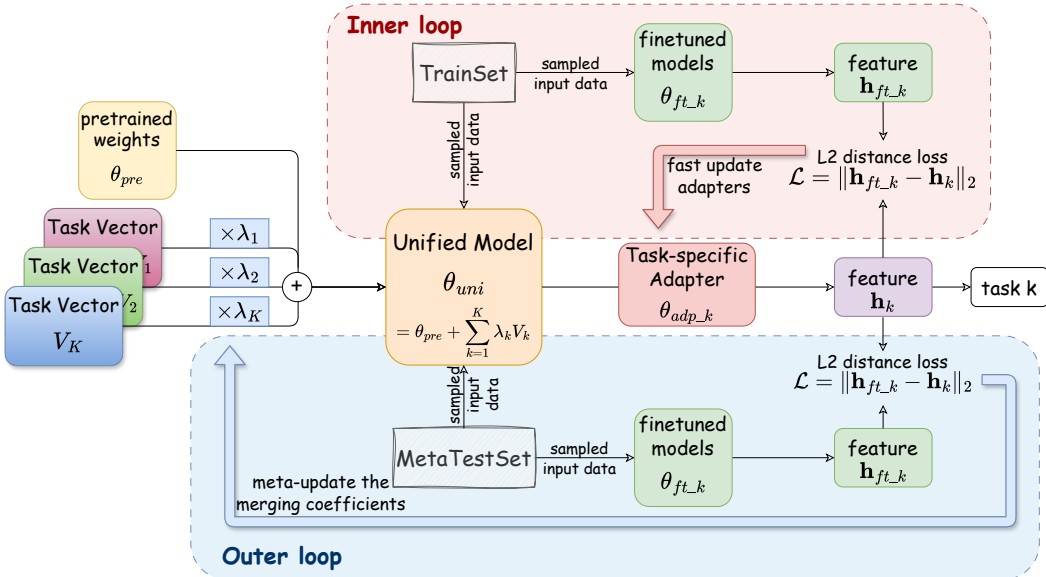

*Figure 2.* The overall framework of MetaMerging consists of two stages: (1) constructing the unified model and (2) training task-specific adapters. The unified model is obtained by merging pretrained parameters and task vectors with coefficients, which are learned via our meta-learning algorithm. In each meta cycle, we first perform a fast update of the adapters and then update the coefficients $\{\lambda_k\}_{k=1}^{K}$ using the meta-gradient. In this way, we leverage episodic adapter training to optimize a better unified model, thereby improving the effectiveness of subsequent adapter training.

*while the static loss satisfies*

$$\mathcal{L}_k^{\text{task}}\left(\theta_{uni}(\lambda)\right) = \mathcal{L}_k^{\text{task}}(\theta_k^\star) + \frac{1}{2}\left\|e_k(\lambda)\right\|_{H_k}^2 + o\left(\left\|e_k(\lambda)\right\|^2\right), \quad (7)$$

*where $P_k$ is the orthogonal projector onto $\mathcal{U}_k$ and $\|v\|_{H_k}^2 = v^\top H_k v$. Consequently, unless a degenerate alignment condition holds (informally, all displacements always lie in all $\mathcal{U}_k$), the minimizers of $F(\lambda)$ and $G(\lambda)$ generally differ:*

$$\arg\min_\lambda F(\lambda) \neq \arg\min_\lambda G(\lambda). \quad (8)$$

The Proof is provided in Appendix A.

**Remark** The theorem shows that downstream adaptation can only correct errors within its constrained update subspace while leaving orthogonal components irreducible. Intuitively, minimizing $F(\lambda)$ penalizes all deviations of $\theta_{uni}(\lambda)$ from each task optimum, including components that adapters can easily recover; in contrast, minimizing $G(\lambda)$ emphasizes reducing the irrecoverable components that cannot be compensated by lightweight adaptation. Therefore, learning $\lambda$ should explicitly account for the downstream adaptation procedure, which naturally yields a *bilevel optimization* problem: $\lambda$ defines the unified model, while adapters are optimized conditioned on that model.

### 3.2. Optimize the Merging Coefficients via Meta-learning

Here, we describe the merging coefficient optimization strategy powered by meta-learning. Recall the research question raised in Section 1, we seek a more generalizable unified model that is easy to adapt to downstream tasks, thus supporting subsequent adapter training. Driven by this, we turn to Meta-learning, which is the process of distilling the experience of multiple learning episodes and using this experience to improve future learning performance (Hospedales et al., 2021). In this paper, our goal is to improve the effectiveness of adapter learning. Therefore, a natural idea is to generate episodes of adapter learning and leverage their experience to improve the adapter learning process itself. In each episode, we use small batches and few updates to rapidly simulate the process of adapter training and testing. And the meta-updated parameters are the merging coefficients used in computing the unified model, which serve as an influential initialization for adapter training.

Specifically, we first split each task dataset into a train set $\mathcal{D}_k$ and a test set $\mathcal{D}_k'$. Within a meta-update cycle, for each task $k$, we sample a batch $\mathbf{X}$ from $\mathcal{D}_k$, and update the adapter parameters $\theta_{ada,k}$ from their initialization by minimizing the alignment loss (Eq. 2) via gradient descent, as follows:

$$\theta_{ada,k}' = \theta_{ada,k} - \alpha \nabla_{\theta_{ada,k}} \mathcal{L}_k(\mathbf{X}; \lambda, \theta_{ada,k}), \quad (9)$$

where $\alpha$ is the step size. For notational simplicity, we consider a single gradient update in this paper, while using multiple gradient updates is a straightforward extension (See Appendix F). Clearly, we can assume that the effectiveness of adapter training is measured by the test performance of $A_{\theta_{ada,k}'}(f_{\theta_{uni}}(\cdot))$ with respect to task $k$. Thus, we further simulate the test process of trained adapters and set the meta-objective to improve the performance across all

tasks. Concretely, we sample a batch input $\mathbf{X}'$ from $\mathcal{D}'_k$ and perform inference using $A_{\theta'_{ada,k}}(f_{\theta_{uni}}(\cdot))$ to get the corresponding features. Similarly, we adopt the feature alignment loss as a metric of test performance, and then perform one-step meta-update on $\{\lambda_k\}_{k=1}^K$ via gradient descent as follows:

$$\lambda \leftarrow \lambda - \beta\nabla_\lambda \sum_{k=1}^K \mathcal{L}_k(\mathbf{X}'; \lambda, \theta'_{ada,k}), \quad (10)$$

where $\beta$ is the meta step size. At this point, a meta-update cycle is completed, and the iteration is repeated until convergence. Thus, we leverage lightweight episodes of adapter training and testing to progressively improve future adapter training. Ultimately, we obtain an optimal $\{\lambda_k\}_{k=1}^K$ and corresponding $\theta_{uni}$, such that small updates of the adapter based on $\theta_{uni}$ can produce large improvements on task performance. Finally, we formulate the bi-level meta-objective as follows:

$$\min_\lambda \sum_{k=1}^K \sum_{\mathbf{X}' \in \mathcal{D}'_k} \mathcal{L}_k(\mathbf{X}'; \lambda, \theta^\star_{ada,k})$$

$$\text{s.t. } \theta^\star_{ada,k}(\lambda) = \arg\min_{\theta_{ada,k}} \sum_{\mathbf{X} \in \mathcal{D}_k} \mathcal{L}_k(\mathbf{X}; \lambda, \theta_{ada,k}), \quad (11)$$

$$k = 1, 2, ..., K,$$

where $\theta^\star_{ada,k}(\lambda)$ represents the optimized adapter parameters given the unified model $\theta_{uni}$. Notably, $\theta_{uni}(\lambda)$ appears inside both Eq. 9 and Eq. 10, which requires computing second-order derivatives for the meta-update of $\{\lambda_k\}_{k=1}^K$ and thus introduces a significant computational burden. To address this, a technique from FOMAML (Nichol et al., 2018) can be applied, which approximates the update by ignoring the second-order terms and using only first-order derivatives, thereby alleviating the computational cost. In this paper, this approximation is employed only for relatively large models such as GPT-2 and ViT-L/14.

Notably, our method is easy to train, as the merging coefficients $\{\lambda_k\}_{k=1}^K$ consist of only $K$ scalar values and converge rapidly in practice. Moreover, the task-specific adapters are lightweight, which makes the overall training process efficient while keeping the final model parameter size compact. Empirical evidence is reported in Table 4. Together with the strong empirical performance demonstrated in our experiments, these results indicate that our method satisfies the criteria of a good model merging approach: efficient and effective. Overall, the workflow of MetaMerging is summarized in Algorithm 1.

### 3.3. Theoretical Analysis from a Regret Perspective

In this section, we provide a theoretical analysis to further elucidate the motivation for our meta-learning framework,

and establish provable guarantees for our method. Recall the objective introduced in Section 1: we adjust the merging coefficient $\lambda$ to facilitate downstream task adaptation. This naturally raises the question: how to formally evaluate the quality of the adapter training process? To this end, we introduce the notion of regret, a theoretical metric widely used in online learning, which naturally characterizes the cumulative optimization error incurred during task-specific adaptation (Khodak et al., 2019; Shalev-Shwartz, 2025). First, we model the adapter training process as an instance of online convex optimization. For task $k$, we consider a sequence of convex loss functions $\{\mathcal{L}_{k,i}(\cdot)\}_{i=1}^m$, each corresponding to a mini-batch encountered during adapter training, where $m$ denotes the total number of training steps. The explicit form of $\mathcal{L}_{k,i}$ is given in Eq. 2. Starting from the initialization, the adapter parameters are updated iteratively using gradient descent on successive mini-batches of data. Let $\theta_{ada,k,i}$ denote the adapter parameters after the $i$-th training step.

We define the task-specific regret of the adapter training process as

$$R_k := \sum_{i=1}^m \mathcal{L}_{k,i}(\theta_{ada,k,i}) - \sum_{i=1}^m \mathcal{L}_{k,i}(\theta^\star_{ada,k}(\lambda)), \quad (12)$$

where $\theta^\star_{ada,k}(\lambda) = \arg\min_\theta \sum_{i=1}^m \mathcal{L}_{k,i}(\theta)$ denotes the optimal task-specific adapter parameter in hindsight, which depends on $\lambda$ through the unified model $\theta_{uni}(\lambda)$ since the training samples $\mathbf{X}$ are fixed for task $k$. The regret measures the cumulative optimization error of the adapted model relative to the best attainable adapter parameter, and can be interpreted as the area between the training loss trajectory and its optimal lower bound. As illustrated in Fig. 5, adapter training under the MetaMerging framework incurs substantially smaller regret than the weight-averaging baseline.

Overall, regret $R_k$ serves as a meaningful metric for evaluating the quality of adapter training: a smaller regret implies both better task performance (lower final loss) and a more efficient process (faster loss reduction). Therefore, our objective is to minimize the regret $R_k$ for all $K$ downstream tasks. Building on classical results in online convex optimization, under standard assumptions of convexity and Lipschitz continuity, the task-specific regret $R_k$ admits an upper bound $U_k$, given by

$$R_k \leq U_k = \frac{1}{2\eta}\left\|\theta^\star_{ada,k}(\lambda)\right\|_2^2 + \eta G_k^2 m, \quad (13)$$

where $\eta$ is the adapter learning rate, and $G_k$ is the Lipschitz constant of the loss functions. We provide the detailed theoretical background of this upper bound in Appendix B. Thus, the objective reduces to minimizing the regret upper bound $U_k$, which directly motivates our method design. Abstracting away the learning rate and other fixed constants, we find that the only dominant term is the $\theta^\star_{ada,k}(\lambda)$, which depends on the merging coefficient $\lambda$. However, manually

*Table 1.* Multi-task performance of merged ViT-B/32 models across eight image classification tasks.

| Methods | SUN397 | Cars | RESISC45 | EuroSAT | SVHN | GTSRB | MNIST | DTD | Avg Acc |
|---|---|---|---|---|---|---|---|---|---|
| Pretrained Model | 62.3 | 59.7 | 60.7 | 45.5 | 31.4 | 32.6 | 48.5 | 43.8 | 48.0 |
| Finetuned Model | 75.3 | 77.7 | 96.1 | 99.7 | 97.5 | 98.7 | 99.7 | 79.4 | 90.5 |
| Multi-task Learning | 73.9 | 74.4 | 93.9 | 98.2 | 95.8 | 98.9 | 99.5 | 77.9 | 88.9 |
| Weight Averaging (Wortsman et al., 2022a) | 65.3 | 63.4 | 71.4 | 71.7 | 64.2 | 52.8 | 87.5 | 50.1 | 65.8 |
| Fisher Merging (Matena & Raffel, 2022) | 68.6 | 69.2 | 70.7 | 66.4 | 72.9 | 51.1 | 87.9 | 59.9 | 68.3 |
| RegMean (Jin et al., 2023) | 65.3 | 63.5 | 75.6 | 78.6 | 78.1 | 67.4 | 93.7 | 52.0 | 71.8 |
| Task Arithmetic (Ilharco et al., 2023) | 63.8 | 62.1 | 72.0 | 77.6 | 74.4 | 65.1 | 94.0 | 52.2 | 70.1 |
| Ties-Merging (Yadav et al., 2023) | 64.8 | 62.9 | 74.3 | 78.9 | 83.1 | 71.4 | 97.6 | 56.2 | 73.6 |
| FR-Merging (Zheng & Wang, 2025) | 66.2 | 64.5 | 77.2 | 90.1 | 85.4 | 82.3 | 98.5 | 60.0 | 78.1 |
| AdaMerging (Yang et al., 2024d) | 64.5 | 68.1 | 79.2 | 93.8 | 87.0 | 91.9 | 97.5 | 59.1 | 80.1 |
| AdaMerging++ (Yang et al., 2024d) | 66.6 | 68.3 | 82.2 | 94.2 | 89.6 | 89.0 | 98.3 | 60.6 | 81.1 |
| Surgery (Yang et al., 2024b) | 69.8 | 71.0 | 88.9 | 98.1 | 91.7 | 96.5 | 98.8 | 73.6 | 86.1 |
| Pareto Merging (Chen & Kwok, 2025) | 72.1 | 73.7 | 88.8 | 97.5 | 92.2 | 97.5 | 99.0 | 66.1 | 85.9 |
| WUDI-Merging (Cheng et al., 2025) | 71.1 | 71.0 | 85.7 | 95.6 | 94.2 | 94.7 | 99.5 | 69.7 | 85.2 |
| DOGE (Wei et al., 2025) | 70.5 | 74.8 | 88.7 | 94.1 | 91.6 | 95.7 | 98.8 | 72.5 | 85.9 |
| **MetaMerging** (Ours) | 74.2 | 71.9 | 92.0 | 99.4 | 97.1 | 98.1 | 99.6 | 64.9 | 87.2 |

*Table 2.* Multi-task performance of merged ViT-L/14 models across eight image classification tasks.

| Methods | SUN397 | Cars | RESISC45 | EuroSAT | SVHN | GTSRB | MNIST | DTD | Avg Acc |
|---|---|---|---|---|---|---|---|---|---|
| Pretrained Model | 66.8 | 77.7 | 71.0 | 59.9 | 58.4 | 50.5 | 76.3 | 55.3 | 64.5 |
| Finetuned Model | 82.3 | 92.4 | 97.4 | 100 | 98.1 | 99.2 | 99.7 | 84.1 | 94.2 |
| Multi-task Learning | 80.8 | 90.6 | 96.3 | 96.3 | 97.6 | 99.1 | 99.6 | 84.4 | 93.5 |
| Weight Averaging (Wortsman et al., 2022a) | 72.1 | 81.6 | 82.6 | 91.9 | 78.2 | 70.7 | 97.1 | 62.8 | 79.6 |
| Fisher Merging (Matena & Raffel, 2022) | 69.2 | 88.6 | 87.5 | 93.5 | 80.6 | 74.8 | 93.3 | 70.0 | 82.2 |
| RegMean (Jin et al., 2023) | 73.3 | 81.8 | 86.1 | 97.0 | 88.0 | 84.2 | 98.5 | 60.8 | 83.7 |
| Task Arithmetic (Ilharco et al., 2023) | 74.1 | 82.1 | 86.7 | 93.8 | 87.9 | 86.8 | 98.9 | 65.6 | 84.5 |
| Ties-Merging (Yadav et al., 2023) | 76.5 | 85.0 | 89.3 | 95.7 | 90.3 | 83.3 | 99.0 | 68.8 | 86.0 |
| FR-Merging (Zheng & Wang, 2025) | 76.4 | 87.0 | 90.2 | 96.8 | 92.0 | 92.8 | 99.3 | 71.5 | 88.3 |
| AdaMerging (Yang et al., 2024d) | 79.0 | 90.3 | 90.8 | 96.2 | 93.4 | 98.0 | 99.0 | 79.9 | 90.8 |
| AdaMerging++ (Yang et al., 2024d) | 79.4 | 90.3 | 91.6 | 97.4 | 93.4 | 97.5 | 99.0 | 79.2 | 91.0 |
| Surgery (Yang et al., 2024b) | 80.3 | 90.8 | 94.3 | 98.2 | 94.1 | 98.7 | 99.2 | 82.5 | 92.3 |
| Pareto Merging (Chen & Kwok, 2025) | 80.6 | 91.7 | 92.0 | 98.5 | 96.1 | 99.0 | 99.1 | 80.6 | 92.2 |
| WUDI-Merging (Cheng et al., 2025) | 81.0 | 91.0 | 94.2 | 99.2 | 96.3 | 98.1 | 99.6 | 81.2 | 92.6 |
| DOGE (Wei et al., 2025) | 79.7 | 91.6 | 94.4 | 96.7 | 96.5 | 98.6 | 99.0 | 84.1 | 92.6 |
| **MetaMerging** (Ours) | 82.1 | 90.6 | 97.2 | 99.7 | 97.9 | 98.9 | 99.7 | 80.9 | 93.4 |

tuning the $\lambda$ is neither practical nor principled. Instead, a rational approach is to learn $\lambda$ via meta-learning.

**Theorem 3.2** (Meta-learning reduces the regret-dominant term). *Assume that each loss $\mathcal{L}_{k,i}$ is convex and $G_k$-Lipschitz. Then, optimizing $\lambda$ via the proposed bi-level meta-learning objective yields a solution $\lambda^\star$ such that*

$$\left\| \theta^\star_{ada,k}(\lambda^\star) \right\|_2^2 \ \leq \ \left\| \theta^\star_{ada,k}(\lambda) \right\|_2^2$$

The proof follows from standard sensitivity analysis of the inner optimization problem and is provided in Appendix B.

## 4. Experiment

In this section, we conduct a series of experiments to thoroughly validate our method. Due to page limitations, we defer the hyperparameter analysis, details of the experimental setup, and additional results to the Appendix.

### 4.1. Merging Vision Models

**Datasets and Models** We follow the most widely adopted model merging setting (Yang et al., 2024d; Huang et al.,

2024; Chen & Kwok, 2024; Shen et al., 2024). Specifically, we employ ViT-B/32 and ViT-L/14, two variants of CLIP (Radford et al., 2021) visual encoders, as the pretrained models. We merge eight finetuned models, each fully finetuned on a downstream image classification task: SUN397 (Xiao et al., 2016), Cars (Krause et al., 2013), RESISC45 (Cheng et al., 2017), EuroSAT (Helber et al., 2019), SVHN (Yuval, 2011), GTSRB (Stallkamp et al., 2011), MNIST (LeCun, 1998), and DTD (Cimpoi et al., 2014). We report the final performance using classification accuracy. Dataset statistics and preprocessing details are provided in the Appendix.

**Baselines** We compare our method with multiple baseline model merging methods, ranging from classic merging methods to recent advanced methods, and the baseline details are provided in Appendix G. These methods fall into two categories: training-free methods (e.g., Weight Averaging (Wortsman et al., 2022a)) and methods requiring additional training (e.g., AdaMerging (Yang et al., 2024d)). Notably, our method produces the final merged model with a parameter size close to a single finetuned model. Thus,

*Table 3.* Multi-task performance of merged GPT-2 models across seven text classification tasks.

| Method | CoLA | SST-2 | MRPC | QQP | MNLI | QNLI | RTE | Avg. |
|---|---|---|---|---|---|---|---|---|
| Pretrained | 30.8 | 50.9 | 31.4 | 63.2 | 33.3 | 49.2 | 52.7 | 44.5 |
| Finetuned | 76.8 | 91.2 | 80.4 | 89.6 | 82.1 | 88.3 | 65.3 | 82.0 |
| Weight Averaging | 55.0 | 52.5 | 51.0 | 76.7 | 55.1 | 57.6 | 44.8 | 56.1 |
| Fisher Merging | 54.8 | 64.7 | 39.5 | 81.5 | 58.0 | 63.3 | 49.1 | 58.7 |
| AdaMerging | 49.7 | 86.7 | 37.5 | 71.2 | 54.8 | 65.3 | 52.1 | 59.6 |
| RegMean | 61.7 | 79.7 | 65.4 | 78.8 | 70.4 | 69.7 | 56.0 | 68.8 |
| Task Arithmetic | 68.7 | 83.6 | 69.6 | 81.8 | 68.6 | 70.5 | 47.3 | 70.0 |
| Ties-Merging | 68.4 | 81.8 | 68.4 | 82.4 | 71.4 | 69.6 | 47.7 | 70.0 |
| **MetaMerging** | 71.8 | 90.8 | 79.4 | 78.8 | 69.0 | 82.5 | 66.2 | 76.9 |

*Table 4.* The storage and computational efficiency of MetaMerging.

| The total number of parameters | |
|---|---|
| ViT-B/32 model | 92,185,089 |
| ViT-L/14 model | 303,179,776 |
| our adapter | 99,136 |
| **Running time on a single RTX 4090 GPU** | |
| Multi-Task Learning | 15h 53m |
| AdaMerging | 2h 5m |
| Surgery | 0h 46m |
| Pareto Merging | 2h 37m |
| MetaMerging | 1h 23m |

here we only compare against methods with a similar parameter scale. For reference, we also evaluate the pretrained model and the individual finetuned models, serving as the upper and lower performance bounds, respectively. Furthermore, we include multi-task learning as a baseline, in which the multi-task model is obtained by jointly training on all downstream datasets (Zhang & Yang, 2021; Vandenhende et al., 2021). To avoid conflating the effect of the adaptation procedure with the quality of the merged initialization, we additionally equip training-free merging baselines with the same task-specific adapters and train them under the identical unlabeled-data budget, optimizer, and training steps, referring to Figure 4 and Table 6 and Table 7.

**Main Results** Table 1 and Table 2 present the multi-task performance of different merging methods on ViT-B/32 and ViT-L/14, respectively. We observe that MetaMerging achieves consistently strong performance across all tasks and model scales. Compared to traditional merging methods without additional training, MetaMerging delivers substantial improvements, often by a margin of 10-20% in average accuracy. Furthermore, even when compared with recent advanced methods that incorporate trainable modules to enhance merging, MetaMerging still surpasses them, demonstrating the effectiveness of our meta-learning based coefficient optimization and adapter training. Moreover, our method achieves performance competitive with multi-task learning without requiring extensive training and access to labeled data.

### 4.2. Merging Language Models

**Datasets and Models** Following the FusionBench benchmark (Tang et al., 2024a), we evaluate model merging in the NLP domain by merging GPT-2 (Radford et al., 2019) models finetuned on seven text classification tasks from GLUE (Wang et al., 2018), which is a popular multi-task benchmark of NLP. Performance on all tasks is measured by accuracy. The dataset statistics and preprocessing details are provided in the Appendix.

**Baselines** We compare our method with several classical baseline model merging methods. Consistent with the experiments on vision models, we also include the pretrained

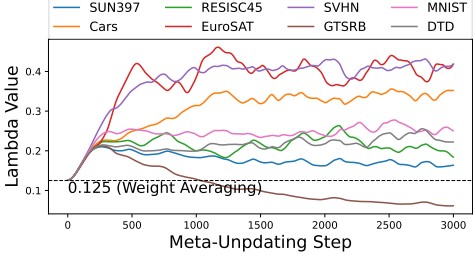

*Figure 3.* Model merging coefficients $\{\lambda_k\}_{k=1}^K$ change with respect to training steps on merging ViT-B/32 models. Each curve represents the change process of the coefficient $\lambda_k$ of a task vector $T_k$ ($k \in \{1, 2, \ldots, K\}$).

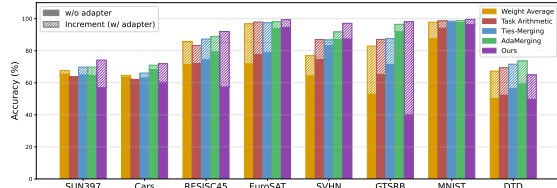

*Figure 4.* Incremental performance gains from task-specific adapters across unified models merged by different methods (merging ViT-B/32 models for 8 vision tasks). Figure 1 in Section 1 presents the average accuracy version of this chart. More results in Table 6 and Table 7 in Appendix

model and finetuned model as references.

**Main Results** Table 3 presents the results of our merging experiment. It can be seen that MetaMerging shows significant performance improvement compared with baseline methods. This demonstrates that our method is applicable not only to vision models but also to language models, and can be extended to various neural network architectures, further verifying the generality and scalability of our method.

### 4.3. Mechanism Analysis of MetaMerging

In this subsection, we aim to investigate the underlying mechanism of MetaMerging and provide insights into how it works. Recall the process of MetaMerging, the core technique is the meta-learning-based optimization of the merging coefficient. The merging coefficient $\{\lambda_k\}_{k=1}^K$ controls the balance across tasks and shapes the unified model, which governs the subsequent adapter training and task performance. Firstly, we conduct experiments by applying the same adapter training procedure to unified models obtained

*Table 5.* Multi-task performance of merged ViT-B/32 models across eight image classification tasks.

| Methods | SUN397 | Cars | RESISC45 | EuroSAT | SVHN | GTSRB | MNIST | DTD | Avg Acc |
|---|---|---|---|---|---|---|---|---|---|
| Finetuned Model | 75.3 | 77.7 | 96.1 | 99.7 | 97.5 | 98.7 | 99.7 | 79.4 | 90.5 |
| WEMOE (Shen et al., 2024) | 74.1 | 77.4 | 93.7 | 99.1 | 96.2 | 98.9 | 99.6 | 76.4 | 89.4 |
| EMR-Merging (Huang et al., 2024) | 75.2 | 72.8 | 93.5 | 99.5 | 96.9 | 98.1 | 99.6 | 74.4 | 88.7 |
| SuperMerge (Yang et al., 2024e) | 67.9 | 73.4 | 90.7 | 97.6 | 95.1 | 96.2 | 97.9 | 66.4 | 85.7 |
| FREE-Merging (Zheng & Wang, 2025) | 77.1 | 78.2 | 93.4 | 99.5 | 96.3 | 98.2 | 99.5 | 75.4 | 89.7 |
| **MetaMerging** (Ours) | 74.2 | 71.9 | 92.0 | 99.4 | 97.1 | 98.1 | 99.6 | 64.9 | 87.2 |

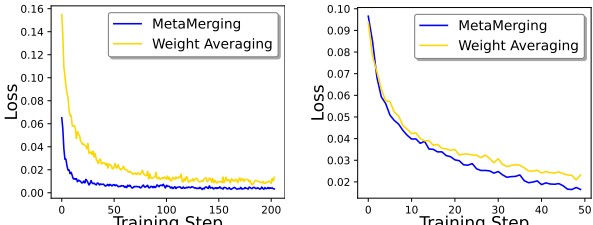

*Figure 5.* Loss curves when training adapters for SVHN and RESISC45 tasks (the other six tasks are reported in the Appendix), based on the unified models obtained by our method and Weight Averaging.

from baseline methods and evaluate their performance, as shown in Figure 4. From the results, we observe that the effectiveness of the additional adapters depends not only on the underlying unified model but also varies significantly across different tasks. This suggests that the characteristics of a task may determine that its performance tends to depend on the unified model or adapters. In other words, for some tasks, most of the knowledge may come from the shared representation, while other tasks may rely more heavily on task-specific knowledge. For instance, in tasks such as GTSRB and RESISC45, the addition of adapters leads to substantial performance improvements, suggesting a significant knowledge gap in the unified model that necessitates task-specific adapters. In contrast, for tasks such as Cars, SVHN, and MNIST, the performance gains from adapters are much smaller, suggesting that the knowledge from task vectors in the unified model is largely sufficient. An intuitive idea is to increase the weight of tasks that do not require adapters and reduce the weight of others, allowing the adapters to focus on compensating for tasks that need additional task-specific knowledge. To investigate whether MetaMerging optimizes the coefficients in this manner, we trace the change process of $\{\lambda_k\}_{k=1}^K$ during the meta-learning procedure, as presented in Figure 3. It can be seen that the coefficients $\lambda$ for Cars, SVHN, and MNIST progressively grow and those for GTSRB and RESISC45 diminish during the procedure, consistent with our hypothesis above. This provides an interpretable insight, showing that our method can automatically adapt to the unique characteristics of each task, thereby achieving a balance and optimal performance across all tasks.

On the other hand, we investigate whether the unified model learned by meta-learning has sufficient generalization capacity to be effectively adapted to downstream tasks. Fig-

ure 5 shows a comparison of the loss curves when training task-specific adapters on the unified model obtained by our method versus Weight Averaging. It can be observed that task-specific adapters trained with our method converge more rapidly and reach lower loss values, even when the initial loss is not particularly low. This indicates that our unified model provides better initialization and greater generalizability for downstream adaptation. Overall, these results further demonstrate both the effectiveness of our method and its interpretability.

### 4.4. Comparison with High-capacity Baselines

To better contextualize the empirical performance of our method, we further compare it with several high-capacity baselines that incur substantially higher training cost, larger task-specific parameter budgets, or extra modules. As shown in Table 5, these methods may achieve slightly higher accuracy, but at orders-of-magnitude higher parameter and storage cost. For 8 tasks, our merged model has 89.05M parameters, which is comparable to a single-task ViT-B/32 (Table 4). However, WEMoE uses 573.96M parameters (Table 8 in (Shen et al., 2024)), i.e., over $6\times$ larger due to its heavier MoE architecture. Moreover, EMR-Merging requires task-specific masks at inference, hindering unified deployment, and FREE-Merging introduces an MoE with a learned router. Despite these substantially increased computational and storage costs, these high-capacity methods only outperform our approach by approximately 1%–2% in task performance and still fall below the task-specific finetuned upper bound. In contrast, our method achieves a more favorable performance–parameter trade-off, maintaining strong empirical performance while preserving high efficiency.

### 4.5. Efficiency Analysis of MetaMerging

We examine the storage and computational costs of MetaMerging. The comparison of parameter counts and runtime is shown in Table 4. The total number of retained parameters consists only of the single unified model and $K$ task-specific adapters. Since the scale of the adapters is negligible compared to the encoder model, our method is storage efficient. Moreover, our meta-learning algorithm finds a good initialization for the unified model, enabling adapter training to converge fast and easily, which is discussed above and verified by our experiments. Empirically,

our adapter training requires only 1 epoch on the validation dataset, whereas multi-task learning typically requires 10 or more epochs. Consequently, we conduct an experiment to compare MetaMerging with baseline methods and multi-task learning, recording the runtime of each merging method.

## 5. Conclusion

In this paper, we address the challenges of recent progressive model merging frameworks. We introduce a multi-task model merging framework, MetaMerging, along with a novel meta-learning algorithm to adaptively optimize the merging coefficients of task vectors. Experimental results demonstrate that our method is effective and intuitively interpretable. Additionally, our method is widely applicable and can be integrated with techniques that focus on processing task vectors, such as sparsifying them or promoting their orthogonality, potentially providing some inspiration for future research on model merging.

## Acknowledgements

The work of Ziyue Qiao was partially supported by the National Natural Science Foundation of China (No. 62406056) and the Guangdong Basic and Applied Basic Research Foundation (No. 2024A1515140114). The work of Qin Zhang was partially supported by the National Natural Science Foundation of China (No. 62576221) and the Guangdong Provincial Natural Science Foundation (No. 2025A1515010288).

## Impact Statement

This paper focuses on advancing the field of machine learning. We do not foresee any societal impacts that require special consideration beyond those commonly associated with progress in this area.

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

# A. Proof of Theorem 3.1

In this appendix, we provide a formal theoretical justification showing that optimizing merging coefficients based on the static performance of the merged model is generally suboptimal when downstream task adaptation is involved.

Given the static merging objective $F(\lambda)$ in Eq. (3) and the post-adaptation objective $G(\lambda)$ in Eq. (2), we assume the following mild conditions hold in a neighborhood of each $\theta_k^\star$.

**(i) Constrained Task Adaptation**   For each task $\tau_k$, downstream adaptation is restricted to an affine subspace

$$\theta \mapsto \theta + u, \quad u \in \mathcal{U}_k, \tag{14}$$

where $\mathcal{U}_k \subset \mathbb{R}^d$ and $\mathcal{U}_k \neq \mathbb{R}^d$ is a proper linear subspace.

**(ii) Local Strong Convexity and Smoothness**   For each task $k$, the task loss $\mathcal{L}_k^{\text{task}}$ is twice continuously differentiable and locally $\mu_k$-strongly convex and $L_k$-smooth in a neighborhood of $\theta_k^\star$. Let

$$H_k = \nabla^2 \mathcal{L}_k^{\text{task}}(\theta_k^\star), \tag{15}$$

with $\mu_k I \preceq H_k \preceq L_k I$.

**(iii) Generic Misalignment**   There exists at least one task $k$ and a coefficient vector $\lambda$ such that the displacement

$$e_k(\lambda) := \theta_{uni}(\lambda) - \theta_k^\star \tag{16}$$

has a non-zero component outside the adaptation subspace $\mathcal{U}_k$.

By condition (ii), in a sufficiently small neighborhood of $\theta_k^\star$, the task loss admits the second-order Taylor expansion

$$\mathcal{L}_k^{\text{task}}(\theta_k^\star + e) = \mathcal{L}_k^{\text{task}}(\theta_k^\star) + \frac{1}{2}e^\top H_k e + o(\|e\|^2). \tag{17}$$

Substituting $e = e_k(\lambda)$ yields the local approximation of the static loss term.

We define the $H_k$-induced inner product $\langle a, b \rangle_{H_k} := a^\top H_k b$ and the corresponding norm $\|v\|_{H_k} := \sqrt{v^\top H_k v}$. Let $P_k$ denote the $H_k$-orthogonal projector onto the adaptation subspace $\mathcal{U}_k$, i.e.,

$$P_k e \in \mathcal{U}_k, \qquad \langle e - P_k e, v \rangle_{H_k} = 0, \quad \forall v \in \mathcal{U}_k. \tag{18}$$

For $\theta_{uni}(\lambda)$ sufficiently close to $\theta_k^\star$, the post-adaptation loss satisfies

$$\min_{u \in \mathcal{U}_k} \mathcal{L}_k^{\text{task}}(\theta_{uni}(\lambda) + u) = \mathcal{L}_k^{\text{task}}(\theta_k^\star) + \frac{1}{2}\min_{u \in \mathcal{U}_k}(e_k(\lambda) + u)^\top H_k(e_k(\lambda) + u) + o(\|e_k(\lambda)\|^2). \tag{19}$$

Ignoring the higher-order term, the inner minimization is a constrained quadratic problem. Its optimal solution $u^\star$ satisfies the first-order optimality condition

$$v^\top H_k\big(e_k(\lambda) + u^\star\big) = 0, \quad \forall v \in \mathcal{U}_k, \tag{20}$$

which implies that the residual $(e_k(\lambda) + u^\star)$ lies in the $H_k$-orthogonal complement of $\mathcal{U}_k$. Consequently,

$$\min_{u \in \mathcal{U}_k}(e_k(\lambda) + u)^\top H_k(e_k(\lambda) + u) = \big\|(I - P_k)e_k(\lambda)\big\|_{H_k}^2. \tag{21}$$

Combining (17) and (21), the local quadratic terms of the static and post-adaptation losses differ by

$$\|e_k(\lambda)\|_{H_k}^2 - \big\|(I - P_k)e_k(\lambda)\big\|_{H_k}^2 = \big\|P_k e_k(\lambda)\big\|_{H_k}^2 \geq 0. \tag{22}$$

This shows that the static objective penalizes both recoverable and irrecoverable components of the task error, whereas the post-adaptation objective depends only on the irrecoverable component.

**Conclusion** From the expansions above, we obtain the local forms

$$\mathcal{L}_k^{\text{task}}(\theta_{uni}(\lambda)) = \mathcal{L}_k^{\text{task}}(\theta_k^\star) + \frac{1}{2}\|e_k(\lambda)\|_{H_k}^2 + o(\|e_k(\lambda)\|^2), \tag{23}$$

$$\min_{u\in\mathcal{U}_k} \mathcal{L}_k^{\text{task}}(\theta_{uni}(\lambda) + u) = \mathcal{L}_k^{\text{task}}(\theta_k^\star) + \frac{1}{2}\|(I - P_k)e_k(\lambda)\|_{H_k}^2 + o(\|e_k(\lambda)\|^2). \tag{24}$$

Unless a degenerate alignment condition holds such that $P_k e_k(\lambda) = 0$ for all tasks $k$ and all coefficients $\lambda$, the quadratic models of $F(\lambda)$ and $G(\lambda)$ are different. Since $\theta_{uni}(\lambda)$ depends affinely on $\lambda$, the induced quadratic objectives generally admit different minimizers. Therefore,

$$\arg\min_\lambda F(\lambda) \neq \arg\min_\lambda G(\lambda), \tag{25}$$

which completes the proof. $\square$

## B. Theoretical background of Regret Upper Bound and Proof of Theorem 3.2.

In this appendix, we present the theoretical background on regret upper bounds and provide the proof of Theorem 3.2. Without loss of generality, we assume that the task-specific adapter is a linear map parameterized by a matrix $W \in \mathbb{R}^{d\times d}$. Under this assumption, for task $k$, the adapter training loss defined in Eq. 2 can be rewritten as

$$\mathcal{L}_{k,i}(W) = \mathcal{L}_k(\mathbf{X}; \lambda, \theta_{ada,k} = \|f_{\theta_k}(\mathbf{X}_i) - A_{\theta_k^{ada}}(f_{\theta_{uni}}(\mathbf{X}_i))\|_2^2 = \|Y_i - WH_i\|_2^2 \tag{26}$$

where $Y_i = f_{\theta_k}(\mathbf{X}_i)$ denotes the task-specific target representation, and $H_i = f_{\theta_{uni}}(\mathbf{X}_i)$ denotes the hidden feature produced by the unified model parameterized by $\theta_{uni}(\lambda)$.

We observe that $\mathcal{L}_{k,i}(W) = \|Y_i - WH_i\|_2^2$ is a quadratic function of $W$, and is therefore convex. Moreover, we assume that $\mathcal{L}_{k,i}(W)$ is $G$-Lipschitz continuous with respect to $W$ over the feasible parameter domain $\mathcal{W}$, i.e.,

$$\|\nabla_W \mathcal{L}_{k,i}(W)\|_F \leq G, \qquad \forall W \in \mathcal{W}. \tag{27}$$

Under these conditions, the task-specific adapter training process can be naturally viewed as an instance of online convex optimization (OCO), where each sample $(H_i, Y_i)$ induces a convex loss revealed sequentially. Without loss of generality, we consider a simple and representative case of OCO, the standard online gradient descent (OGD), as the within-task adapter learning algorithm. We denote $W_1$ as the initialization of the adapter parameters, $\eta > 0$ as the step size, and $m$ as the total number of update rounds. At each round $i$ of the adapter parameter update, after observing the batch loss $\mathcal{L}_{k,i}(\cdot)$, OGD performs

$$W_{i+1} = W_i - \eta\nabla\mathcal{L}_{k,i}(W_i), \tag{28}$$

For each task $k$, define the comparator (the optimal action in hindsight)

$$W_k^\star \in \arg\min_{W\in\mathcal{W}} \sum_{i=1}^{m_k} \mathcal{L}_{k,i}(W), \tag{29}$$

and the within-task regret after $m$ rounds as

$$\text{Regret}_k(W_0, \eta) := \sum_{i=1}^{m_k} \mathcal{L}_{k,i}(W_i) - \sum_{i=1}^{m_k} \mathcal{L}_{k,i}(W_k^\star). \tag{30}$$

**A Standard Regret Upper Bound** Under the definition above, according to previous study on OCO (Eq. 1 in (Khodak et al., 2019), theorem 2.11 in (Shalev-Shwartz, 2025)), this training process admits a classical and well-known regret bound:

$$\text{Regret}_k(W_1, \eta) \leq U_k(W_1, \eta) := \frac{1}{2\eta}\|W_k^\star - W_1\|_2^2 + \eta\sum_{i=1}^{m_k}\|\nabla\mathcal{L}_{k,i}(W_{k,i})\|_2^2 \leq \frac{1}{2\eta}\|W_k^\star - W_1\|_2^2 + \eta G^2 m_k. \tag{31}$$

We emphasize the strong data dependence through $W_k^\star$, which depends on the entire task-$k$ sequence $\{\mathcal{L}_{k,i}\}_{i=1}^{m_k}$ and finally depends on the model merging coefficient $\lambda$.

Here, we assume that the initialization is set to $W_1 = \mathbf{0}$, which is consistent with our empirical setting. Substituting $W_1 = \mathbf{0}$ into the above bound yields

$$U_k = \frac{1}{2\eta}\|W_k^\star\|_2^2 + \eta G^2 m_k, \tag{32}$$

which corresponds to Eq. 13 in Section 3.

We now proceed to the proof of the Theorem 3.2, which provides a theoretical justification for the effectiveness of our meta-learning strategy. Note that the merging coefficient $\lambda$ only affects the adapter training process through the unified representations $H_i = f_{\theta_{uni}(\lambda)}(\mathbf{X}_i)$. Consequently, proving Theorem 3.2 reduces to showing that the representations $H_i$ learned by our bi-level meta-learning objective yield a smaller value of $\|W_k^\star\|_F^2$.

To make the analysis explicit, we consider the simplest learning scenario, in which the parameters reaches its optimal solution in a single update step, i.e., the total number of update steps is set to 1.

$$W_k^\star = W - \eta\nabla_W\mathcal{L}_{k,i}(W), \qquad H^\star = H - \gamma\nabla_H\mathcal{L}_{meta}(H) \tag{33}$$

Notably, for notational simplicity, we omit the task and iteration subscripts $k$ and $i$, and directly use $W$, $H$, $Y$, and $\mathcal{L}(\cdot)$ to denote the corresponding initial quantities, where the adapter is initialized as $W = \mathbf{0}$.

We first perform one inner gradient descent step on $W$ to obtain a temporary iterate $W'$. Using standard matrix calculus,

$$\nabla_W\mathcal{L}(W, H) = -2(Y - WH)H^\top = 2(WH - Y)H^\top = -2EH^\top. \tag{34}$$

where $E := Y - WH$. Then one gradient-descent update gives

$$W' = W - \eta\nabla_W L(W, H) = W - 2\eta(WH - Y)H^\top = W + 2\eta EH^\top. \tag{35}$$

Plug $W'$ back into the loss, we get

$$\begin{aligned} Y - W'H &= Y - \left(W - 2\eta(WH - Y)H^\top\right)H \\ &= Y - WH + 2\eta(WH - Y)H^\top H \\ &= E - 2\eta EH^\top H \\ &= E\left(I - 2\eta H^\top H\right). \end{aligned} \tag{36}$$

Therefore, the one-step post-update loss (the meta-level loss for optimizing $H$) is

$$\mathcal{L}_{meta}(H) = L(W', H) = \left\|Y - W'H\right\|_2^2 = \left\|E\left(I - 2\eta H^\top H\right)\right\|_2^2. \tag{37}$$

We further calculate the gradient of $\mathcal{L}_{meta}(H)$ w.r.t. $H$. Let

$$S := H^\top H, \qquad A := I - 2\eta S. \tag{38}$$

Then $\mathcal{L}_{meta}(H) = \|EA\|_F^2 = \operatorname{tr}(A^\top E^\top EA)$ with $A^\top = A$. By the chain rule for matrix-valued functions, we have

$$\nabla_H\mathcal{L}_{meta} = \left(\frac{\partial E}{\partial H}\right)^* \nabla_E\mathcal{L}_{meta} + \left(\frac{\partial A}{\partial H}\right)^* \nabla_A\mathcal{L}_{meta}.$$

Thus, the computation decomposes into two terms.

**(I)** Since $dE = -W\,dH$ and for fixed $A$ we have $\mathcal{L}_{meta} = \|EA\|_F^2$,

$$\begin{aligned} d\mathcal{L}_{meta} &= 2\langle EA, dE\,A\rangle_F = 2\langle EA, (-W\,dH)A\rangle_F \\ &= -2\langle W^\top EAA^\top, dH\rangle_F, \end{aligned} \tag{39}$$

thus

$$\left(\frac{\partial E}{\partial H}\right)^* \nabla_E\mathcal{L}_{meta} = -2W^\top EA^2 = -2W^\top E(I - 2\eta S)^2. \tag{40}$$

**(II)** For fixed $E$, $\mathcal{L}_{meta} = \operatorname{tr}(AE^\top EA) = \operatorname{tr}(AMA)$ where $M := E^\top E$. Hence

$$
\begin{aligned}
d\mathcal{L}_{meta} &= \operatorname{tr}(dA\, MA) + \operatorname{tr}(A\, M\, dA) \\
&= \operatorname{tr}\big(dA(MA + AM)\big).
\end{aligned}
\tag{41}
$$

Since $A = I - 2\eta H^\top H$, we have

$$
dA = -2\eta\,(dH^\top H + H^\top dH).
\tag{42}
$$

Let $B := MA + AM$ (note $B$ is symmetric because $M$ and $A$ are symmetric). Then

$$
\begin{aligned}
d\mathcal{L}_{meta} &= \operatorname{tr}\big(-2\eta(dH^\top H + H^\top dH)B\big) \\
&= -2\eta\operatorname{tr}(dH^\top HB) - 2\eta\operatorname{tr}(H^\top dH\,B) \\
&= -2\eta\operatorname{tr}(BH^\top dH) - 2\eta\operatorname{tr}(BH^\top dH) \\
&= -4\eta\operatorname{tr}(BH^\top dH) = -4\eta\,\langle HB,\, dH\rangle_F.
\end{aligned}
\tag{43}
$$

Therefore,

$$
\left(\frac{\partial A}{\partial H}\right)^{*}\nabla_A\mathcal{L}_{meta}. = -4\eta\, H\,(MA + AM) = -4\eta\, H\Big(E^\top E(I - 2\eta S) + (I - 2\eta S)E^\top E\Big).
\tag{44}
$$

Combining (40) and (44), we obtain the final meta-gradient

$$
\nabla_H\mathcal{L}_{meta}(H) = -2W^\top E(I - 2\eta S)^2 - 4\eta\, H\Big(E^\top E(I - 2\eta S) + (I - 2\eta S)E^\top E\Big),
\tag{45}
$$

**Meta-update on $H$** With meta learning rate $\gamma > 0$, we get (substituting $W = 0$)

$$
H^\star = H - \gamma\nabla_H\mathcal{L}_{meta}(H) = H + 4\gamma\eta\, H\Big(Y^\top Y(I - 2\eta S) + (I - 2\eta S)Y^\top Y\Big),
\tag{46}
$$

After achieving the meta-learned $H^\star$, we perform inner radient descent step on W again to get

$$
W^\star = W - \eta\nabla_W L(W, H^\star).
\tag{47}
$$

We now compare $\|W'\|_2^2$ with $\|W^\star\|_2^2$, which correspond to the adapter parameters before and after the meta update, respectively. When $W = 0$, we have

$$
\|W'\|_2^2 = \|\nabla_W\mathcal{L}(W, H)\|_2^2 = \|2YH^\top\|_2^2.
\tag{48}
$$

After the meta update, we obtain

$$
\|W^\star\|_2^2 = \|\nabla_W\mathcal{L}(W, H^\star)\|_2^2 = \|2Y(H^\star)^\top\|_2^2.
\tag{49}
$$

Since $H = H^\star + \Delta H$ with $\Delta H$ aligned with $Y^\top Y \succeq 0$,

$$
\langle YH^\top,\, Y(\Delta H)^\top\rangle_F = 4\gamma\eta\operatorname{tr}\Big(Y^\top Y\big(Y^\top Y(I - 2\eta H^\top H) + (I - 2\eta H^\top H)Y^\top Y\big)H^\top H\Big) \geq 0,
\tag{50}
$$

it follows that

$$
\|YH^\top\|_F^2 = \|Y(H^\star)^\top\|_F^2 + 2\langle Y(H^\star)^\top,\, Y(\Delta H)^\top\rangle_F + \|Y(\Delta H)^\top\|_F^2 \geq \|Y(H^\star)^\top\|_F^2,
\tag{51}
$$

Therefore, after meta-optimizing the representation $H$ to the optimal solution $H^\star$, the dominant term in the objective is reduced, i.e.,

$$
\|W^\star\|_2^2 \leq \|W'\|_2^2.
\tag{52}
$$

Under the notation of Section 3, the above result corresponds to

$$
\big\|\theta_{ada,k}^\star(\lambda^\star)\big\|_2^2 \leq \big\|\theta_{ada,k}^\star(\lambda)\big\|_2^2,
\tag{53}
$$

which completes the proof of Theorem 3.2. $\qquad\square$

# C. Overall Algorithmic Pipeline of MetaMerging.

---

**Algorithm 1** MetaMerging

---

**Require:** fine-tuned model $\{f_{\theta_1}, f_{\theta_2}, ..., f_{\theta_K}\}$, pretrained model $f_{\theta_{pre}}$, unlabeled datasets $\mathcal{D}_k$, initialized task-specific adapters $\{A_{\theta_{ada\_1}}, A_{\theta_{ada\_2}}, ..., A_{\theta_{ada,k}}\}$, Merging coefficients $\{\lambda_k\}_{k=1}^K$, loss fuction $\mathcal{L}$ (L2 distance), step size hyperparameters $\alpha$, $\beta$

1:
2: **Coarsely merging:**
3: Compute the unified model $f_{\theta_{uni}}$:
4:     $\theta_{uni}(\lambda) = \theta_{pre} + \sum_{k=1}^K \lambda_k(\theta_k - \theta_{pre})$
5:
6: **Meta-learning merging coefficients:**
7: **while** not done **do**
8:     **for** each task $k$ **do**
9:         Sample inputs $\mathbf{X}_i$ from train dataset $\mathcal{D}_k$
10:         Compute alignment loss: $\mathcal{L}_k = \|f_{\theta_k}(\mathbf{X}_i) - A_{\theta_{ada,k}}(f_{\theta_{uni}}(\mathbf{X}_i))\|_2$
11:         Perform a one-step gradient update of the task-specific adapter from its initialization:
12:         $\theta'_{ada,k} = \theta_{ada,k} - \alpha\nabla_{\theta_{ada,k}}\mathcal{L}_k$
13:         Sample test inputs $\mathbf{X}'_i$ from meta-test dataset $\mathcal{D}'_k$ for upper-level mate-update
14:         Compute the test metric as meta loss: $\mathcal{L}'_k = \|f_{\theta_k}(\mathbf{X}'_i) - A_{\theta'_{ada,k}}(f_{\theta_{uni}}(\mathbf{X}'_i))\|_2$,
15:         imitating the test-time inference of trained adapters on downstream task.
16:     **end for**
17:     Update $\{\lambda_k\}_{k=1}^K$ with gradient descent: $\lambda \leftarrow \lambda - \beta\nabla_\lambda \sum_{k=1}^K \mathcal{L}'_k$
18: **end while**
19:
20: **Training Adapters:**
21: **for** each task $k$ **do**
22:     **while** not convergent **do**
23:         Sample inputs $\mathbf{X}_i$ from train datasets $\mathcal{D}_k$
24:         Train the task-specific adapter $\theta_{ada,k}$ from initialization,
25:         by minimizing the loss: $\mathcal{L}_k = \|f_{\theta_k}(\mathbf{X}_i) - A_{\theta_{ada,k}}(f_{\theta_{uni}}(\mathbf{X}_i))\|_2$
26:     **end while**
27: **end for**
28:
**Ensure:** Obtain optimal $\theta_{uni}$ with respect to $\{\lambda_k\}_{k=1}^K$ and final trained $\{\theta_{ada\_1}, \theta_{ada\_2}, ..., \theta_{ada,k}\}$

---

# D. Additional Results

Here, we provide the additional results as a supplement to the experiments in the main text. Table 6 and Table 7 present the detailed results of the incremental performance analysis in Section 4.3, where the same adapter training process is applied to unified models obtained by different merging methods. Figure 6 shows loss curves of adapter training on the other 6 image classification tasks, complementing the results in Figure 5.

*Table 6.* Multi-task performance when merging ViT-B/32 models on eight tasks.

| Methods | SUN397 | Cars | RESISC45 | EuroSAT | SVHN | GTSRB | MNIST | DTD | Avg Acc |
|---|---|---|---|---|---|---|---|---|---|
| Pretrained model | 62.3 | 59.7 | 60.7 | 45.5 | 31.4 | 32.6 | 48.5 | 43.8 | 48.0 |
| Finetuned model | 75.3 | 77.7 | 96.1 | 99.7 | 97.5 | 98.7 | 99.7 | 79.4 | 90.5 |
| Weight Averaging w/o adapters | 65.3 | 63.4 | 71.4 | 71.7 | 64.2 | 52.8 | 87.5 | 50.1 | 65.8 |
| Task Arithmetic w/o adapters | 63.8 | 62.1 | 72.0 | 77.6 | 74.4 | 65.1 | 94.0 | 52.2 | 70.1 |
| Ties-Merging w/o adapters | 64.8 | 62.9 | 74.3 | 78.9 | 83.1 | 71.4 | 97.6 | 56.2 | 73.6 |
| AdaMerging w/o adapters | 64.5 | 68.1 | 79.2 | 93.8 | 87.0 | 91.9 | 97.5 | 59.1 | 80.1 |
| **MetaMerging** w/o adapters | 56.8 | 60.3 | 57.3 | 94.5 | 87.2 | 40.2 | 96.3 | 49.6 | 67.8 |
| Weight Averaging w/ adapters | 67.6 | 64.6 | 85.8 | 96.8 | 76.9 | 82.9 | 97.8 | 67.3 | 80.0 |
| Task Arithmetic w/ adapters | 63.8 | 59.9 | 83.3 | 97.9 | 87.0 | 87.0 | 98.6 | 69.4 | 80.9 |
| Ties-Merging w/ adapters | 69.8 | 66.1 | 87.3 | 97.5 | 86.7 | 87.6 | 98.5 | 71.6 | 83.1 |
| AdaMerging w/ adapters | 69.8 | 71.0 | 88.9 | 98.1 | 91.7 | 96.5 | 98.8 | 73.6 | 86.1 |
| **MetaMerging** | 74.2 | 71.9 | 92.0 | 99.4 | 97.1 | 98.1 | 99.6 | 64.9 | 87.2 |

*Table 7.* Multi-task performance when merging ViT-L/14 models on eight tasks.

| Methods | SUN397 | Cars | RESISC45 | EuroSAT | SVHN | GTSRB | MNIST | DTD | Avg Acc |
|---|---|---|---|---|---|---|---|---|---|
| Pretrained model | 66.8 | 77.7 | 71.0 | 59.9 | 58.4 | 50.5 | 76.3 | 55.3 | 64.5 |
| Finetuned model | 82.3 | 92.4 | 97.4 | 100 | 98.1 | 99.2 | 99.7 | 84.1 | 94.2 |
| Weight Averaging w/o adapters | 72.1 | 81.6 | 82.6 | 91.9 | 78.2 | 70.7 | 97.1 | 62.8 | 79.6 |
| Task Arithmetic w/o adapters | 74.1 | 82.1 | 86.7 | 93.8 | 87.9 | 86.8 | 98.9 | 65.6 | 84.5 |
| Ties-Merging w/o adapters | 76.5 | 85.0 | 89.3 | 95.7 | 90.3 | 83.3 | 99.0 | 68.8 | 86.0 |
| AdaMerging w/o adapters | 79.0 | 90.3 | 90.8 | 96.2 | 93.4 | 98.0 | 99.0 | 79.9 | 90.8 |
| **MetaMerging** w/o adapters | 74.7 | 80.1 | 86.9 | 92.8 | 88.6 | 87.2 | 98.9 | 65.9 | 84.4 |
| Weight Averaging w/ adapters | 73.7 | 83.9 | 92.0 | 98.4 | 82.4 | 86.3 | 98.7 | 71.9 | 85.9 |
| Task Arithmetic w/ adapters | 75.7 | 84.4 | 93.1 | 98.8 | 91.3 | 93.4 | 99.1 | 76.1 | 89.0 |
| Ties-Merging w/ adapters | 76.5 | 85.9 | 93.7 | 99.2 | 89.7 | 92.0 | 99.1 | 78.1 | 89.3 |
| AdaMerging w/ adapters | 80.3 | 90.8 | 94.3 | 98.2 | 94.1 | 98.7 | 99.2 | 82.5 | 92.3 |
| **MetaMerging** | 82.1 | 90.6 | 97.2 | 99.7 | 97.9 | 98.9 | 99.7 | 80.9 | 93.4 |

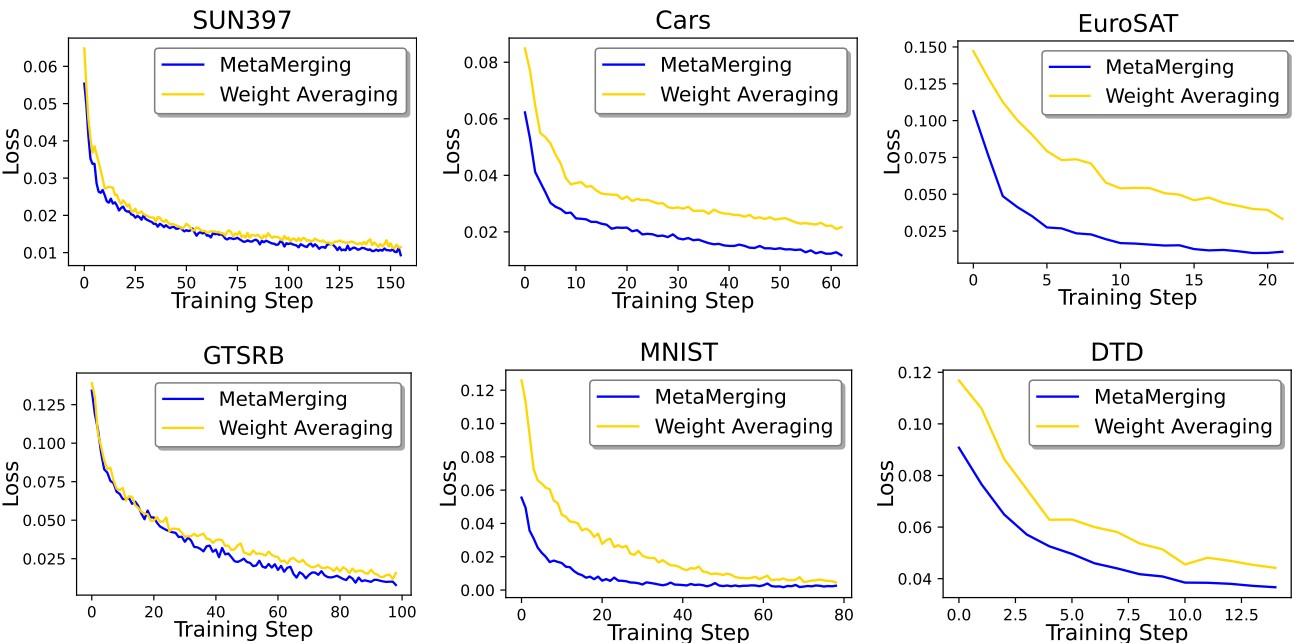

*Figure 6.* The detailed results of experiments in Figure 5 Section 4.3. We present the additional loss curve of adapter training on 6 tasks.

# E. Experiment Setup Details

**Data Preprocessing and Splitting**   Following the (Ilharco et al., 2023) and (Huang et al., 2024) for vision and language model merging experiments, we split each dataset into training, validation, and test sets. In our meta-learning algorithm (Algorithm 1), we use the training set as $\mathcal{D}_k$, the validation set as the mate-test dataset $\mathcal{D}'_k$, and the test set as the final evaluation of the performance of MetaMerging. The data preprocessing strategy also follows the previous work (Ilharco et al., 2023).

**Training Setting and Computational Environment**   In our experiments, we adopt an adapter architecture as follows

$$A_\theta(\mathbf{h}) = \mathbf{h} - Linear(ReLU(Linear(h)))$$

where $Linear(\mathbf{h}) = \mathbf{W} \cdot \mathbf{h} + \mathbf{b}$ is a simple linear layer, and the hidden dimension inside the adapter, referred to as the rank, determines the scale of the adapter. In all experiments, we set the adapter rank to 64, while the token feature dimension is 768. We set inner step size $\alpha$ to 0.1 and $\beta$ to 0.01 and perform one gradient update in the inner loop. The sensitivity of these hyperparameters is analyzed in Appendix F. For merging ViT-B/32 models, we run the experiments on a single NVIDIA RTX 4090 GPU; for merging ViT-L/14 and GPT-2 models, we run the experiments in parallel on 4 NVIDIA RTX 4090 GPUs. For the runtime experiments, the adapter is trained for one epoch in stage (2) of our method, and for multi-task learning, it is trained for ten epochs.

## F. Hyper-parameter Analysis

We further examine the impact of hyper-parameters (step size $\alpha$ and meta step size $\beta$) in our meta-learning algorithm. As shown in Table 8 and Table 9, we present the performance variation of our method with $\alpha$ and $\beta$ ranging from 1 to 0.001, respectively. It can be observed that too large or too small values of both $\alpha$ and $\beta$ lead to suboptimal performance. By the way, we record the runtime and compare it to multi-task learning.

Additionally, in the meta-learning iteration of our method, the number of inner-step updates for the adapter is a hyperparameter that plays a critical role in the efficiency of the meta-learning process. We evaluate its impact when merging GPT-2 models and report the runtime and average performance in Figure 7 and Table 10.

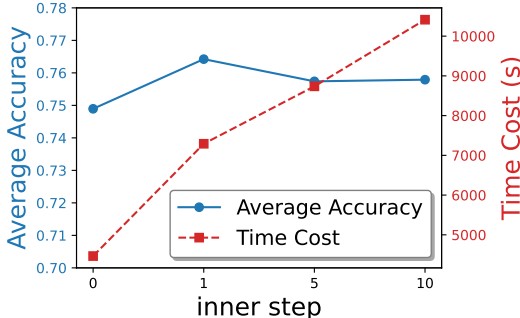

*Figure 7.* Impact of inner steps when merging GPT-2 models

*Table 8.* Impact of different step sizes $\alpha$ in meta-learning. Multi-task performance of merging ViT-B/32 models on eight image classification tasks. Training time is measured on a single NVIDIA RTX 4090 GPU.

| Methods | SUN397 | Cars | RESISC45 | EuroSAT | SVHN | GTSRB | MNIST | DTD | Avg Acc | training times |
|---|---|---|---|---|---|---|---|---|---|---|
| Pretrained Model | 62.3 | 59.7 | 60.7 | 45.5 | 31.4 | 32.6 | 48.5 | 43.8 | 48.0 | None |
| Finetuned Model | 75.3 | 77.7 | 96.1 | 99.7 | 97.5 | 98.7 | 99.7 | 79.4 | 90.5 | None |
| Multi-task Learning | 73.9 | 74.4 | 93.9 | 98.2 | 95.8 | 98.9 | 99.5 | 77.9 | 88.9 | 15h 53m 16s |
| step size 1 | 73.53 | 70.00 | 92.57 | 99.52 | 97.31 | 98.53 | 99.50 | 62.98 | 86.74 | 1h 36m 52s |
| step size 0.1 | 74.34 | 70.46 | 92.44 | 99.52 | 97.36 | 97.72 | 99.57 | 64.84 | 87.15 | 1h 42m 58s |
| step size 0.01 | 74.55 | 70.40 | 92.51 | 99.52 | 97.31 | 97.14 | 99.54 | 63.83 | 86.85 | 1h 35m 27s |
| step size 0.001 | 74.45 | 70.25 | 92.56 | 99.52 | 97.34 | 97.40 | 99.53 | 63.78 | 86.85 | 1h 37m 16s |

*Table 9.* Impact of different meta step sizes $\beta$ in meta-learning. Multi-task performance of merging ViT-B/32 models on eight image classification tasks. Training time is measured on a single NVIDIA RTX 4090 GPU.

| Methods | SUN397 | Cars | RESISC45 | EuroSAT | SVHN | GTSRB | MNIST | DTD | Avg Acc | training times |
|---|---|---|---|---|---|---|---|---|---|---|
| Pretrained Model | 62.3 | 59.7 | 60.7 | 45.5 | 31.4 | 32.6 | 48.5 | 43.8 | 48.0 | None |
| Finetuned Model | 75.3 | 77.7 | 96.1 | 99.7 | 97.5 | 98.7 | 99.7 | 79.4 | 90.5 | None |
| Multi-task Learning | 73.9 | 74.4 | 93.9 | 98.2 | 95.8 | 98.9 | 99.5 | 77.9 | 88.9 | 15h 53m 16s |
| meta step size 1 | 72.72 | 65.75 | 94.98 | 99.59 | 97.36 | 96.66 | 99.36 | 59.52 | 85.74 | 1h 29m 42s |
| meta step size 0.1 | 74.39 | 70.66 | 91.44 | 99.56 | 97.37 | 97.69 | 99.52 | 66.38 | 87.13 | 1h 31m 34s |
| meta step size 0.01 | 74.34 | 70.46 | 92.44 | 99.52 | 97.36 | 97.72 | 99.57 | 64.84 | 87.15 | 1h 23m 34s |
| meta step size 0.001 | 74.46 | 69.03 | 93.11 | 99.30 | 97.26 | 98.60 | 99.50 | 62.18 | 86.68 | 1h 57m 4s |

*Table 10.* Impact of different inner steps in meta-learning. Multi-task performance when merging GPT-2 models on seven text classification tasks. The training time is measured on distributed training with 4 NVIDIA RTX 4090 GPUs.

| Method | CoLA | SST-2 | MRPC | QQP | MNLI | QNLI | RTE | Avg ACC | training time (s) |
|---|---|---|---|---|---|---|---|---|---|
| Pretrained model | 30.8 | 50.9 | 31.4 | 63.2 | 33.3 | 49.2 | 52.7 | 44.5 | None |
| Finetuned model | 76.8 | 91.2 | 80.4 | 89.6 | 82.1 | 88.3 | 65.3 | 82.0 | None |
| inner steps 0 | 69.3 | 89.6 | 76.2 | 78.2 | 60.1 | 84.7 | 66.2 | 74.9 | 6465.49 |
| inner steps 1 | 71.5 | 88.4 | 78.9 | 78.8 | 67.5 | 84.0 | 65.8 | 76.4 | 7291.45 |
| inner steps 5 | 69.5 | 90.8 | 78.4 | 82.9 | 61.9 | 80.1 | 66.5 | 75.7 | 8736.94 |
| inner steps 10 | 71.4 | 88.3 | 79.4 | 82.8 | 63.0 | 81.3 | 64.4 | 75.8 | 10413.92 |

# G. Baseline Details

We provide a brief introduction of the baseline method used in our model merging experiments as follows:

**Weight Averaging** (Wortsman et al., 2022a): A simple ensembling approach that averages the parameters of finetuned models. While computationally efficient, it often leads to performance degradation due to conflicting task-specific knowledge.

**Fisher Merging** (Matena & Raffel, 2022): Utilizes the Fisher information matrix to weight model parameters during merging, aiming to preserve important knowledge from each model.

**RegMean** (Jin et al., 2023): Introduces a regularized averaging strategy to mitigate conflicts between models and achieve smoother parameter interpolation.

**Task Arithmetic** (Ilharco et al., 2023): Constructs task vectors from finetuned models and merges them through linear arithmetic with coefficients.

**Ties-Merging** (Yadav et al., 2023): Improves merging stability by resolving parameter conflicts based on tied weights across models.

**AdaMerging** (Yang et al., 2024d): Learns adaptive merging coefficients via lightweight optimization, enabling task-aware parameter integration.

**AdaMerging++** (Yang et al., 2024d): An enhanced version of AdaMerging with refined strategies for coefficient learning and conflict mitigation.

**Surgery** (Yang et al., 2024b): Add task-specific surgery modules (adapters) on unified model to mitigate the "representation bias" of model merging, thereby narrowing the performance gap between model merging and individual finetuned models.

**Pareto Merging** (Chen & Kwok, 2025): Treat model merging as a multi-task optimization problem. By introducing parameter-efficient structures, it generates a Pareto set of merged models, from which users can select according to their preferences or tailor to a specific task.

**WEMoE** (Shen et al., 2024): converts task-conflicting layers into a Mixture-of-Experts while statically merging the remaining parameters, enabling input-dependent expert routing to mitigate interference across tasks.

**EMR-Merging** (Huang et al., 2024): selects a unified base model and applies lightweight task-specific masks and rescaling factors to align parameter directions and magnitudes, achieving strong data-free multi-task merging.

**WUDI-Merging** (Cheng et al., 2025): exploits the low-dimensional linear structure of task vectors to suppress destructive interference during merging, enabling robust data-free consolidation without tuning merging coefficients.

**DOGE** (Wei et al., 2025): formulates model merging as a constrained optimization problem and performs adaptive projected gradient descent in a shared subspace to learn task-aware merging coefficients without relying on training data.

**SuperMerge** (Yang et al., 2024e): leverages gradient-based signals to estimate layer-wise model importance, allowing multiple finetuned models to be systematically merged into a single model with minimal additional training.

**FREE-Merging** (Zheng & Wang, 2025): performs frequency-domain filtering of model parameters to remove task-specific noise and combines it with lightweight experts at inference time, achieving a favorable performance–efficiency trade-off.

# H. Dataset Details

Here, we summarize the details of the datasets used in our experiments. All datasets used in our experiments are publicly accessible.

**Image Classification Datasets**  are illustrated in Table 11. All images are resized to 224×224 for ViT input. Standard data augmentations are applied (random crop, horizontal flip, normalization).

*Table 11.* Image Classification Datasets for ViT Merge Experiments

| Dataset | #Train | #Test | #Classes | Image Size / Notes |
|---|---|---|---|---|
| SUN397 | 108,754 | 8,000 | 397 | Natural scenes, resized to 224×224 |
| Cars (Stanford Cars) | 8,144 | 8,041 | 196 | Car models, resized to 224×224 |
| RESISC45 | 22,500 | 9,000 | 45 | Remote sensing images, resized to 224×224 |
| EuroSAT | 21,600 | 5,400 | 10 | Satellite images, resized to 224×224 |
| SVHN | 73,257 | 26,032 | 10 | Street view house numbers, resized to 32×32 → 224×224 |
| GTSRB | 39,209 | 12,630 | 43 | Traffic signs, resized to 32–250×32–250 → 224×224 |
| MNIST | 60,000 | 10,000 | 10 | Handwritten digits, resized to 28×28 → 224×224 |
| DTD | 3,760 | 1,880 | 47 | Texture dataset, resized to 224×224 |

**Text Classification Datasets**  are illustrated in Table 12. We tokenize all datasets with the GPT-2 tokenizer and pad tokens to the maximum sequence length.

*Table 12.* Text Classification Datasets for GPT-2 Merge Experiments

| Dataset | #Train | #Validation | #Test | Task Type |
|---|---|---|---|---|
| CoLA | 8,551 | 1,043 | 1,063 | Linguistic acceptability (binary) |
| SST-2 | 67,349 | 872 | 1,821 | Sentiment analysis (binary) |
| MRPC | 3,668 | 408 | 1,725 | Paraphrase detection (binary) |
| QQP | 363,849 | 40,430 | 391,315 | Question similarity (binary) |
| MNLI | 392,702 | 9,815 / 9,824 | 9,815 / 9,824 | Natural language inference (3-class) |
| QNLI | 104,743 | 5,463 | 5,463 | Question-answer entailment (binary) |
| RTE | 2,490 | 277 | 3,000 | Textual entailment (binary) |
| WNLI | 634 | 71 | 146 | Coreference / entailment (binary) |

