# OpenReview forum: "Learn to Merge: Meta-Learning for Adaptive Multi-Task Model Merging"
_ICML.cc/2026/Conference — ICML 2026 regular_

### Official Review · Reviewer_kfu6 · 2026-02-16

**Soundness:** 3
**Presentation:** 3
**Significance:** 3
**Originality:** 4
**Overall Recommendation:** 5
**Confidence:** 4

**Summary:**

The method proposes a novel meta-learning framework for optimizing merging coefficients in the scenario of adapter learning, where each task has an adapter. The authors prove that optimizing merging coefficients in static merging is suboptimal in adapter learning and use unsupervised learning to optimize both the merging coefficients and the adapters. Experiments are conducted with relatively small models on standard tasks spanning image classification and GLUE.

**Compliance With Llm Reviewing Policy:**

Affirmed.

**Final Justification:**

I did not have any concern about the contribution of the work itself; the weaknesses and questions I pointed out are pretty much all about clarifications. The rebuttal was to the point and clarified all my doubts, so my final rating is still very positive.

**Key Questions For Authors:**

1. What constraints or regularizations are imposed on the merging coefficients λ (e.g., simplex constraint, non-negativity, normalization across layers)? Do learned λ ever become negative, and how sensitive are results to such choices?

2. How are Dk and D’k formed for the meta-episodes? Are they from validation splits only (no labels used), and are they disjoint at the batch level within an episode? How many samples per episode and how many meta-episodes are typically used?

3. How many adapter parameters per task are used in vision/NLP (architectural details, insertion points), and are they identical across tasks? Do larger adapters further reduce the performance gap to fine-tuned models?

4. Can MetaMerging be combined with layer-wise adaptive methods (e.g., AdaMerging/AdaRank) by using your meta-objective to tune their per-layer λ/masks? Any preliminary results?

5. What is the effect of reducing unlabeled data availability for meta-episodes (few-shot unlabeled regimes)? Is there a minimum episode size for stable coefficient updates?

6. Could you share the λ values per task (mean ± std) and discuss interpretability patterns beyond the few highlighted datasets? For example, what datasets consistently receive large merging coefficients? What makes their coefficients large?

**Limitations:**

While MetaMerging is a timely and sound contribution, extending the meta-learning framework to layer-wise merging coefficients, nonconvex regret analysis, and diverse tasks beyond classification (e.g., generation, many-task regimes) would further demonstrate its generality.

**Strengths And Weaknesses:**

## Strengths

1. The idea of optimizing for post-adaptation performance is theoretically supported and well motivated.
2. The preliminary merging step in MetaMerging is compatible with many existing merging methods, providing a promising direction that is orthogonal to many prior lines of improvement.
3. Clear comparisons with both training-free and training-based merging methods; also includes a discussion against higher-capacity router/MoE-style approaches with parameter/compute trade-offs.
4. The methodological narrative is coherent and easy to follow: from motivation and bilevel formulation to the concrete instantiation (feature alignment loss), and a concise inner/outer loop description.
5. The mention of regret is well-suited; the novelty lies in using regret as an analytical lens to justify meta-learning the merging coefficients.




## Weaknesses
1. The theoretical results rest on strong simplifying assumptions (local strong convexity around per-task optima, linear adapter and convexity/Lipschitz assumptions, one-step updates). While common in meta-learning theory, the connection to overparameterized deep networks and the specific nonlinear adapters used in practice remains largely heuristic.
2. The learned coefficients are global. Authors do not explore finer-grained setups like layer-wise or module-wise granularity.
3. Experimental gaps or methodological issues
4. Lacking details on the train-test split.
5. Lacking some references. MetaMerging is an iterative optimization framework; some other works have explored iterative model merging ([1,2])


[1] Merging Beyond: Streaming LLM Updates via Activation-Guided Rotations: https://arxiv.org/pdf/2602.03237

[2] ATM: Improving Model Merging by Alternating Tuning and Merging: https://arxiv.org/abs/2411.03055

---

> ### Author Rebuttal · Authors · 2026-03-31
>
> Thank you for the thoughtful feedback. We provide our detailed response as follows. For the relevant references, please kindly refer to our rebuttal to **Reviewer hasu**.
>
> **W1**
>
> We agree that our theoretical analysis relies on simplifying assumptions (e.g., local convexity and one-step updates), which are standard in meta-learning theory. Our goal is not to fully characterize the behavior of overparameterized deep networks, but to provide intuition on why optimizing the merging coefficients $\lambda$ for post-adaptation performance is beneficial. Importantly, the analysis highlights the key mechanism of our method: aligning the unified model with downstream adaptation, rather than modeling all practical nonlinearities. Despite these simplifications, our empirical results consistently validate the theoretical insights across diverse architectures and tasks.
>
> **W2**
>
> We further conduct experiments to study different parameterizations of the merging coefficients $\lambda$, including task-wise, layer-wise, and module-wise designs. The results are shown below.
>
> |Method|SUN397|Cars|RESISC45|EuroSAT|SVHN|GTSRB|MNIST|DTD|Avg.|
> |-|-|-|-|-|-|-|-|-|-|
> |**task-wise $\lambda$ (original)**|74.2|71.9|92.0|99.4|97.1|98.1|99.6|64.9|87.2|
> |**layer-wise $\lambda$**|73.8|73.6|92.9|99.3|96.9|99.0|99.6|59.0|86.8|
> |**module-wise $\lambda$**|74.4|75.5|93.4|99.4|97.0|98.8|99.6|70.9|88.6|
>
> We observe that the module-wise parameterization achieves the best performance, suggesting that finer-grained control over parameter merging can further improve results. However, this comes at the cost of increased computational overhead and a larger number of meta-parameters. Therefore, we adopt the task-wise parameterization in our main experiments as a more efficient and scalable trade-off.
>
> **W4**
>
> Following the setting in [6], we randomly split each task dataset into training, validation, and test sets with proportions of 0.8, 0.1, and 0.1, respectively. The training and validation sets are used for meta-training and meta-testing, respectively, i.e. $\mathcal{D}_k$ and $\mathcal{D}'_k$. After meta-learning, we combine them for final adapter training, while the test set is used for final evaluation.
>
> **W5**
>
> Thank you for pointing out these relevant works. We will include the suggested references in the revised version. While these works also explore iterative model merging, our method differs in that MetaMerging formulates the process as a bilevel meta-learning problem, where the merging coefficients are optimized to improve post-adaptation performance. We will clarify these distinctions and provide a more comprehensive discussion in the related work section.
>
> **Q1**
>
> In our implementation, we apply a softplus transformation to the merging coefficients $\lambda$, which ensures that all coefficients are strictly non-negative. We do not impose additional constraints (e.g., simplex normalization), allowing flexible weighting across tasks. Empirically, we observe stable training behavior and consistent performance across different runs, suggesting that the method is not overly sensitive to this design choice.
>
> **Q2**
>
> Please refer to **W4** for details on the dataset split. In our main experiments, we sample 8 instances per task (i.e., $8\times K$ samples per meta-episode), and use 3000 meta-episodes to train the merging coefficients $\lambda$.
>
> **Q3**
>
> The total number of adapter parameters used is 99,136 when merging ViT models. We report the parameter count and analyze the efficiency of our method in Table 4 (Section 4.5). Empirically, larger adapters can further reduce the performance gap to fine-tuned models, but at the cost of increased GPU memory usage and training time.
>
> **Q4**
>
> Please refer to **W2**.
>
> **Q5**
>
> In our implementation, we set the inner-loop batch size to 8 and the outer-loop batch size to 4, indicating that our method does not rely on large amounts of unlabeled data per meta-episode. Larger meta-batch sizes mainly increase computational cost, while moderate sizes are sufficient for stable updates. Regarding the minimum episode size, we have not conducted a systematic study due to time constraints, and leave this as future work.
>
> **Q6**
>
> Here, we run five random seeds for our main experiments and report the meta-learned $\lambda$ value for each task (mean ± std) as follows:
>
> |task|SUN397|Cars|RESISC45|EuroSAT|SVHN|GTSRB|MNIST|DTD|
> |-|-|-|-|-|-|-|-|-|
> |$\lambda$ (mean ± std)|0.17742 ± 0.00061|0.36652 ± 0.00065|0.16304 ± 0.00034|0.48354 ± 0.00139|0.44542 ± 0.00031|0.04858 ± 0.00011|0.23684 ± 0.00140|0.23876 ± 0.00075|
>
> Along with the detailed $\lambda$ trajectories shown in Figure 3 and our analysis in Section 4.1, we observe that datasets which benefit more from task-specific adapters tend to receive smaller $\lambda$ values, likely due to their distinctive data patterns. Conversely, datasets that are more similar with other tasks like Cars and EuroSAT tend to receive larger $\lambda$ values.

---

> > ### Author Rebuttal · Reviewer_kfu6 · 2026-03-31
> >
> > Many thanks to the authors for providing these clarifications.
> >
> > My concerns were mostly on clarifications, and the rebuttal convinced me. I will maintain my positive recommendation.

---

> > > ### Author Response · Authors · 2026-04-01
> > >
> > > We sincerely thank the reviewer for the positive feedback and for acknowledging our clarifications.
> > >
> > > We are glad that our responses have addressed your concerns. We will further improve the clarity of the paper and incorporate the discussed details in the final version.
> > >
> > > Thank you again for your support.

---

### Official Review · Reviewer_FozQ · 2026-03-09

**Soundness:** 3
**Presentation:** 3
**Significance:** 3
**Originality:** 3
**Overall Recommendation:** 5
**Confidence:** 4

**Summary:**

This paper addresses multi-task model merging by linearly interpolating their task vectors with learnable coefficients. The key observation is that existing methods optimize coefficients to maximize the static performance, rather than the performance obtained after training task-specific adapters. The authors formally show that these two objectives are generally misaligned. To address this mismatch, the paper proposes a MAML-style bi-level optimization framework, MetaMerging, that explicitly simulate the adapter training process. Furthermore, the paper also provides a regret-based theoretical analysis to show that the learned coefficients reduce the upper bound on optimization error. Experiments on CV and NLP fields demonstrate that MetaMerging outperforms existing merging methods.

**Compliance With Llm Reviewing Policy:**

Affirmed.

**Final Justification:**

The authors have addressed my main concerns. And I raise my score.

**Key Questions For Authors:**

Given that the search space consists of only K scalar coefficients (typically <20), could the authors compare the proposed meta-learning approach against simpler black-box optimization baselines (e.g., Bayesian optimization or evolutionary strategies) that directly optimize post-adaptation performance over $\lambda$? Such a comparison would help clarify whether the observed improvements stem primarily from optimizing a better objective or from the specific meta-learning optimization procedure.

**Limitations:**

The authors have discussed the societal impact but do not discuss the limitations of the work.

**Strengths And Weaknesses:**

Strength:

 1. The paper formally establishes that the static merging objective is generally misaligned with the post-adaptation objective. While this discrepancy may appear intuitive in hindsight, it had not been formally articulated in prior work.
 2. The optimization is conducted over only K scalar merging coefficients, making the search space extremely low-dimensional. This leads to fast convergence (within ~1000 meta-update steps), minimal computational overhead, and ease of implementation. The method introduces negligible additional cost.
 3. The experimental section is well organized and effectively supports the theoretical claims.

Weakness:

  1. The central argument of the paper is that static merging optimizes a proxy that is misaligned with the true post-adaptation objective. However, the proposed meta-learning procedure introduces its own chain of approximations. While some of these are understandable practical choices, such as using feature alignment in place of downstream task loss and employing first-order meta-gradients to reduce computational cost, the use of a single gradient step to simulate full adapter training is a more fundamental concern. The merging coefficients are optimized to benefit a one-step adaptation, yet at deployment time the adapters are trained to convergence over many steps. A single gradient step captures only local curvature information and may not reflect the outcome of full adapter training, yet no analysis is provided to characterize this discrepancy.
  2. Unlike data-free merging approaches such as Task Arithmetic and TIES-Merging, the proposed method requires access to task-specific training and validation data, as well as forward passes through all fine-tuned models during meta-training. This partially undermines a key practical motivation for model merging, namely the setting where original training data is unavailable or individual model access is limited after merging.

---

> ### Author Rebuttal · Authors · 2026-03-31
>
> Thank you for the thoughtful feedback. We provide our detailed response as follows. For the relevant references, please kindly refer to our rebuttal to **Reviewer hasu**.
>
> **W1**
>
> We thank the reviewer for this insightful comment. We want to clarify that in our setting, the gap is less severe than in standard meta-learning scenarios that involve training an entire model from scratch. Specifically, our downstream adaptation only trains lightweight task-specific adapters on top of a unified model that already contains substantial task knowledge after merging, rather than optimizing the full model from random initialization. Therefore, the downstream adaptation is a computationally efficient process. In fact, in our experiments, the adapters are finally trained for only one epoch on each task after meta-learning procedure, which makes the practical gap between the one-step simulation and the actual adaptation procedure much smaller.
>
> Moreover, the role of the one-step update is not to reproduce the entire optimization trajectory, but to expose how the merging coefficients $\lambda$ affect the early adaptation behavior of the adapters. Since $\lambda$ determines the quality of the unified initialization, even a one-step update can already indicate whether it gives the adapters a better starting point for fast training. In this sense, the one-step objective provides a tractable yet adaptation-aware signal for selecting $\lambda$. In addition, the empirical improvement of MetaMerging on the final post-adaptation performance suggests that the meta-learned coefficients generalize beyond the one-step proxy and remain effective under the actual multi-step adaptation.
>
> We will add more analysis in the revised version.
>
> **W2**
>
> As model merging research evolves, many advanced methods have introduced additional training procedures and leveraged task-specific unlabeled data input to achieve improved merging performance. For example, AdaMerging [2] learns coefficients via entropy maximization, Surgery [3] trains task-specific adapters to align representations, and Twin-Merging [8] trains a router for dynamic parameter combination.
>
> Our approaches operate under the same setting to these work, with the assumption that incorporating lightweight modules and limited additional training, as well as accessing unlabeled task-specific data, is practically acceptable in real-world scenarios. Our goal is to make a trade-off between performance and efficiency, where modest additional computation and lightweight modules lead to improved merging performance.
>
> **Q1**
>
> We further implement two black-box optimization baselines, **Random Search** and **Evolutionary Strategy**, which directly optimize the post-adaptation performance over the merging coefficients $\lambda$. Specifically, for every candidate $\lambda$, we first construct the unified model $\theta_{\mathrm{uni}}(\lambda)=\theta_{\mathrm{pre}}+\sum_{k=1}^{K}\lambda_k V_k$, then train task-specific adapters from scratch under the same setting in our main experiment, and use the averaged post-adaptation performance across tasks as the black-box objective.
>
> For Random Search, we randomly sample 30 candidate $\lambda$ from the search space $[0,2]^k$ (following the setting in Task Arithmetic [6]), evaluate each candidate independently, and report the best-performing one.
>
> For Evolutionary Strategy, we use a simple population-based evolutionary strategy with the same total evaluation budget of 30 candidate $\lambda$. We initialize the population by uniformly sampling 6 candidate coefficient vectors from $[0,2]^k$. At each generation, we evaluate all candidates and select the top 2 performing $\lambda$ as parents. We then generate the next population by applying Gaussian perturbations to the parent $\lambda$:
> $$
> \lambda'=\text{clip}(\lambda + \epsilon,\, 0,\, 2),
> \qquad \epsilon \sim \mathcal{N}(0, \sigma^2 I).
> $$
>
> where $\sigma = 0.1$. Concretely, each parent produces 3 offspring, yielding 6 candidates per generation. We repeat this process for 5 generations and report the best-performing candidate.
>
> We evaluate all methods on merging ViT-B/32 models, and the results are shown below. Due to time and computational constraints, we did not use a larger number of candidate $\lambda$, but we believe the current evaluation budget is sufficient for a meaningful comparison.
> |Method|SUN397|Cars|RESISC45|EuroSAT|SVHN|GTSRB|MNIST|DTD|Avg.|
> |-|-|-|-|-|-|-|-|-|-|
> |**Random Search**|64.8|67.1|78.2|94.5|88.0|90.7|96.8|60.1|80.0|
> |**Evolutionary Strategy**|68.8|71.0|87.9|97.3|91.3|95.5|98.8|72.6|85.4|
> |**MetaMerging**|74.2|71.9|92.0|99.4|97.1|98.1|99.6|64.9|87.2|
>
> The results show that MetaMerging outperforms both Random Search and Evolutionary Strategy across most tasks and achieves the best average performance. This suggests that the gains of our method are not only due to the post-adaptation objective, but also from the proposed bilevel meta-learning optimization procedure.

---

> > ### Author Rebuttal · Reviewer_FozQ · 2026-04-03
> >
> > I thank the authors for the detailed response.
> >
> > The new experiments on Q1 are convincing, and the clarification on W2 is reasonable.
> > However, my concern on W1 remains. The argument that the approximation gap is small is asserted rather than demonstrated; no analysis, qualitative or quantitative, is provided to support this claim. I will largely keep my score.

---

> > > ### Author Response · Authors · 2026-04-03
> > >
> > > Thank you for the follow-up. We have conducted experiments on the number of inner-loop steps, as shown in Figure 7 and Table 10 in Appendix F. We are sorry for not mentioning this in our original rebuttal. The results show that increasing the number of inner steps does not lead to significant performance gains, suggesting that the one-step approximation is empirically sufficient. At the same time, using more inner steps leads to considerable additional computational overhead. This suggests that the one-step inner update is already sufficient in our framework. Therefore, we use one inner step in the main experiments as the best trade-off. We hope this empirical evidence would alleviate your concern. We will revise the paper to highlight this empirical evidence more clearly.

---

### Official Review · Reviewer_hcdx · 2026-03-11

**Soundness:** 4
**Presentation:** 3
**Significance:** 3
**Originality:** 4
**Overall Recommendation:** 5
**Confidence:** 3

**Summary:**

This paper studies multi-task model merging and focuses on how to construct a unified model that is not only good after static merging, but also well suited for subsequent task-specific adaptation. The paper proposes a meta-learning-based framework that learns the merging coefficients through a bilevel optimization process. Specifically, the paper meta-learns task-vector merging coefficients through bilevel optimization and shows that adaptation-aware merging differs from static merging both theoretically and empirically. The method is evaluated on both vision and language benchmarks, including CLIP-based image classification tasks and GPT-2-based GLUE tasks, and is shown to outperform several prior model merging baselines.

**Compliance With Llm Reviewing Policy:**

Affirmed.

**Key Questions For Authors:**

**Q1**: The paper assumes access to training data, whereas model merging is often an alternative paradigm under data isolation constraints arising from privacy, regulation, or data access restrictions. If the data used to train the expert models are already available, then alternative paradigms such as model distillation or multi-task fine-tuning may appear more natural than model merging. The authors should better clarify the problem scenario and discuss why model merging is the most appropriate formulation under their assumptions.

**Q2:** Since the proposed method is explicitly optimized for post-merging adaptation, is it entirely fair to compare it against baselines that are mainly designed for static merging without adaptation-aware modifications? The paper should clarify whether all methods are compared under the same adaptation budget, training steps, and data access assumptions. Otherwise, it is difficult to tell whether the gains come from the proposed method itself or simply from allowing a stronger adaptation procedure. In particular, it would not be surprising if an adaptation-based method outperformed a purely static merging baseline.

**Q3**: The proposed method is built around a MAML-like bilevel design, with inner-loop adapter updates and outer-loop meta-optimization of the merging coefficients. Could the authors provide ablations that separately remove or simplify the inner loop and the outer loop, to verify that both are necessary? In addition, is there a risk of outer-loop memorization or meta-overfitting, where the learned coefficients become overly specialized to the meta-training tasks rather than transferable to unseen tasks?

**Limitations:**

yes

**Strengths And Weaknesses:**

**Strengths**

**S1**: The paper is overall clearly written and easy to understand.

**S2**: The key idea is well motivated. Instead of optimizing merging coefficients only for immediate post-merging performance, the paper explicitly considers post-adaptation performance and formulates the problem in a MAML-like bilevel optimization framework.

**S3**: The method shows consistent improvements over several prior model merging baselines. The experimental evaluation covers both vision and language benchmarks, which gives solid support for the generality of the method.

**Weaknesses**

**W1**: My main concern is that the practical scenario targeted by this model merging setting is not sufficiently clear. Please see details in **Q1**.

**W2:** The baseline selection might not be fair, although I am not fully certain about this point. Please see details in **Q2**.

**W3:** The paper lacks a comprehensive ablation study to convincingly validate the proposed MAML-like mechanism. Please see **Q3** for more details.

---

> ### Author Rebuttal · Authors · 2026-03-31
>
> Thank you for the thoughtful feedback. We provide our detailed response as follows. For the relevant references, please kindly refer to our rebuttal to **Reviewer hasu**.
>
> **Q1**
>
> We would like to clarify that our method requires training that accesses task-specific data inputs, but not ground-truth labels. This unlabeled-data-only setting has been widely adopted in many model merging works with additional training [2][3][4]. We believe this assumption is practically reasonable, as unlabeled data is typically easy to obtain in real-world scenarios. Accordingly, multi-task fine-tuning is not applicable in our setting, as it requires access to labeled data for all tasks. As for model distillation, distilling knowledge from multiple task-specific models into a single multi-task model can be computationally expensive, and the performance may be suboptimal, especially when covering diverse tasks. In contrast, model merging provides a more efficient alternative by directly combining task knowledge at the parameter level. Any additional training involved (if required) is lightweight compared to full multi-task training or large-scale distillation, making it a more practical solution. We will further clarify the problem setting and its practical assumptions in the revised version of our paper.
>
> **Q2**
>
> In Figure 1a (Section 1) and Tables 6 and 7 (Appendix D), we compare our method with static merging baselines equipped with adapters. The adapter training procedure is conducted under the same settings as our method to ensure a fair comparison. From the results, we conclude that MetaMerging achieves superior post-adaptation performance, as it meta-learns more effective merging coefficients that provide a better initialization for downstream adaptation. In addition, we also compare our method with training-based baselines (e.g., AdaMerging and ParetoMerging) in our main experiments, where our method consistently outperforms them.
>
> **Q3**
>
> We further conduct ablation studies to analyze the effectiveness of each component. Specifically, we consider two variants: (1) **w/o inner-loop**: we directly optimize $\lambda$ to convergence, then perform downstream adaptation; (2) **w/o out-loop**: we use the initialized value ($\lambda_k = 1/K$) to construct the unified model and perform downstream adaptation directly. The results are shown below:
>
> | Method | SUN397 | Cars | RESISC45 | EuroSAT | SVHN | GTSRB | MNIST | DTD | Avg. |
> |-|-|-|-|-|-|-|-|-|-|
> | **w/o inner-loop** | 69.5 | 71.2 | 87.9 | 98.1 | 91.5 | 96.5 | 98.1 | 73.4 | 85.8 |
> | **w/o outer-loop** | 67.6 | 64.6 | 85.8 | 96.8 | 76.9 | 82.9 | 97.8 | 67.3 | 80.0 |
> | **MetaMerging** | 74.2 | 71.9 | 92.0 | 99.4 | 97.1 | 98.1 | 99.6 | 64.9 | 87.2 |
>
> We observe that performance degrades when either the inner loop or the outer loop is removed, indicating that both components are essential. This validates the necessity of the proposed bilevel meta-learning framework.
>
> We additionally conduct experiments to investigate whether the merging coefficient suffer from meta-overfitting to meta-training tasks. Specifically, in the ViT-B/32 merging setup, we apply the meta-learning procedure and construct a unified model from 6 downstream tasks. We then perform subsequent adaptation and evaluation on 2 unseen tasks to evaluate the generalizability of meta-learned unified model. Following the setting in [2], we compare our method with several baselines, including Task Arithmetic [6], Ties-Merging [7], AdaMerging [2], and AdaMerging++ [2]. We evaluate under two configurations, where the unseen tasks are MNIST & EuroSAT and RESISC45 & SVHN, respectively. The results are summarized below:
>
> |Method|SUN397|Cars|RESISC45|DTD|SVHN|GTSRB|Avg.|*UNSEEN:*|MNIST|EuroSAT|Avg.|
> |-|-|-|-|-|-|-|-|-|-|-|-|
> |Task Arithmetic [6]|63.3|62.4|75.1|57.8|84.6|80.4|70.6||77.2|46.2|61.7|
> |Ties-Merging [7]|67.8|66.2|77.2|56.7|77.1|70.9|69.3||75.9|43.3|59.6|
> |AdaMerging [2]|65.2|65.9|88.5|61.1|92.2|91.5|77.4||84.0|56.1|70.0|
> |AdaMerging++ [2]|68.2|67.6|86.3|63.6|92.6|89.8|78.0||83.9|53.5|68.7|
> |**MetaMerging**|73.9|69.3|94.2|69.0|97.4|98.7|83.7||99.5|94.4|96.9|
>
> |Method|SUN397|Cars|GTSRB|EuroSAT|DTD|MNIST|Avg.|*UNSEEN:*|RESISC45|SVHN|Avg.|
> |-|-|-|-|-|-|-|-|-|-|-|-|
> |Task Arithmetic [6]|64.0|64.0|75.2|87.7|57.0|95.7|73.9||52.3|44.9|51.1|
> |Ties-Merging [7]|68.0|67.1|67.7|78.4|56.5|92.8|71.8||58.7|49.2|53.9|
> |AdaMerging [2]|67.1|67.8|94.8|94.4|59.6|98.2|80.3||50.2|60.9|55.5|
> |AdaMerging++ [2]|68.9|69.6|91.6|94.3|61.9|98.7|80.8||52.0|64.9|58.5|
> |**MetaMerging**|73.2|72.4|98.3|99.6|65.1|99.6|84.7||88.7|96.6|92.6|
>
> The results show that MetaMerging achieves the best performance in both two configurations, maintaining effective adaptation on both seen and unseen tasks. This indicates that the meta-learned coefficients do not overfit to the training tasks and exhibit transferability.

---

### Official Review · Reviewer_hasu · 2026-03-13

**Soundness:** 3
**Presentation:** 3
**Significance:** 3
**Originality:** 3
**Overall Recommendation:** 4
**Confidence:** 5

**Summary:**

This paper proposes MetaMerging, a novel meta-learning framework for adaptive multi-task model merging. The core idea is to learn optimal merging coefficients that produce a unified model which serves as a superior initialization for subsequent task-specific adapter training. The motivation—to optimize the merged model not for its immediate performance but for its adaptability—is well-articulated and addresses a genuine limitation in existing static merging methods. The theoretical analysis provides formal justification for this perspective, and the method design is intuitive.

**Compliance With Llm Reviewing Policy:**

Affirmed.

**Key Questions For Authors:**

Please see the Weaknesses above

**Limitations:**

No. The author pointed out the performance shortcomings of MetaMerging compared to high-capacity baselines. However, there is no clear discussion of the limitations, particularly the performance degradation of the w/o adapter merged model, which is a key concern.

**Strengths And Weaknesses:**

### **Paper Strengths**
1. This paper correctly identifies a key deviation in the current model merging paradigm: optimizing the static performance of the unified model is usually suboptimal, which provides new insights for future work.
2. The paper provides clear evidence and a solid theoretical foundation for this work.
3. This paper is well organized, and the figures are understandable.

### **Paper Weaknesses**
1. Comparing Table 1 and Figure 4, it is evident that the baseline results reported in Table 1 correspond to a unified model without any adapters. In contrast, MetaMerging trains a separate adapter for each dataset. This comparison is therefore unfair. Moreover, as shown in Figure 4, MetaMerging without adapters generally underperforms the baseline methods. Does this indicate a failure of the merging step? It appears that the merged model has learned biased shared features.
2. The original motivation behind model merging is to achieve Pareto optimality across multiple tasks with the merged model. However, introducing a separate adapter for each task seems to contradict this goal, as it prevents the model from treating all tasks equally. If such a paradigm is desired, why not simply adopt a pre-trained model with LoRA fine-tuning, which would avoid the complex merging procedure altogether?
3. The paper employs a feature alignment loss as the inner-loop objective in meta-learning, rather than task-specific losses such as MSE or cross-entropy. What advantages does this choice offer? The paper does not provide sufficient explanation or analysis regarding this design decision.
If author can address my concern, I will raise my score.

---

> ### Author Rebuttal · Authors · 2026-03-31
>
> Thank you for the thoughtful feedback. We provide our detailed response as follows.
>
> **W1**
>
> In our main experiments (Tables 1 and 2), we follow the original settings of each baseline in their respective papers. We categorize them into two groups: training-free methods (e.g., Weight Averaging [1]) and training-required methods (e.g., AdaMerging [2]), separated by a black line in the tables. Accordingly, training-free methods do not include adapters, while training-required methods introduce additional trainable modules (e.g., adapters in Surgery [3] and LoRAs in Pareto Merging [4]). we will clarify this more explicitly in the revised version.
>
> More importantly, comparisons with baselines that involve adapters are already presented in Figure 1a and in Tables 6 and 7. In addition, As stated in Section 1 Lines 55–58:
>
> > we found the optimal unified model is not necessarily the one that performs best on all tasks, but the one that better preserves the potential for adapters, thereby enabling greater downstream gains.
>
> our meta-learning objective is not to learn a unified model with the best immediate performance, but to learn a strong initialization tailored for downstream adapter training, leading to more balanced and effective multi-task performance. Therefore, the lower performance of MetaMerging without adapters is not a failure merging step, but a better initialization that yields superior post-adaptation performance.
>
> **W2**
>
> As model merging research evolves, introducing additional task-specific modules has become a common strategy in many prior works [3][4][5]. This does not contradict the motivation of model merging, since only a small fraction of parameters is task-specific, while the vast majority remains shared across tasks, thereby reducing storage and computational overhead.
>
> Our setting is motivated by practical scenarios where multiple task-specific models are already available, and the goal is to efficiently consolidate their knowledge into a unified model that can be further adapted to each task with lightweight modules. In this context, directly applying LoRA to the pretrained model is not a strong alternative. A well-constructed unified model produced by model merging already captures multi-task knowledge, serving as a better starting point for downstream adaptation.
>
> We further support this point with experiments on merging ViT-B/32. We compare pretrained models with LoRA/adapters and different unified models with adapters. The results are shown below:
>
> |Method|SUN397|Cars|RESISC45|EuroSAT|SVHN|GTSRB|MNIST|DTD|Avg.|
> |-|-|-|-|-|-|-|-|-|-|
> |**pretrainedmodel + LoRAs**|68.0|60.3|80.5|84.8|93.9|88.2|98.4|47.6|77.7|
> |**pretrainedmodel + adapters**|69.0|59.0|81.6|85.7|94.4|88.0|98.6|46.1|77.8|
> |**unifiedmodel (WeightAveraging) + adapters**|67.6|64.6|85.8|96.8|76.9|82.9|97.8|67.3|80.0|
> |**unifiedmodel (AdaMerging) + adapters**|69.8|71.0|88.9|98.1|91.7|96.5|98.8|73.6|86.1|
> |**unifiedmodel (MetaMerging) + adapters**|74.2|71.9|92.0|99.4|97.1|98.1|99.6|64.9|87.2|
>
> We observe that directly applying LoRA/adapters on a pretrained model often leads to suboptimal performance compared with applying adapters on an effectively merged unified model.
>
> **W3**
>
> We would like to clarify that the MSE or cross-entropy mentioned by the reviewer are typical supervised losses, which require access to task-specific labels. However, such labels are often unavailable in real-world model merging scenarios. In contrast, our alignment loss can be viewed as an MSE defined on the outputs (or representations) of fine-tuned models, i.e., a form of knowledge distillation. Since fine-tuned models typically achieve strong task-specific performance, using them as soft targets provides an effective supervision signal without requiring actual labels. Therefore, our method is both effective and practical. Additionally, these setting that only access to unlabeled data is widely adopted in many model merging works with additional training [2][3][4].
>
> [1]ICML'2022 Model soups: averaging weights of multiple fine-tuned models improves accuracy without increasing inference time. https://arxiv.org/abs/2203.05482
> [2]ICLR'2024 AdaMerging: Adaptive Model Merging for Multi-Task Learning. https://openreview.net/forum?id=nZP6NgD3QY
> [3]ICML'2024 Representation Surgery for Multi-Task Model Merging. https://openreview.net/forum?id=Sbl2keQEML
> [4]ICML'2025 Pareto Merging: Multi-Objective Optimization for Preference-Aware Model Merging. https://openreview.net/forum?id=D7qRwx6BOS
> [5]NeurIPS'2024 EMR-Merging: Tuning-Free High-Performance Model Merging. https://openreview.net/forum?id=lYdjzx3DYu
> [6]ICLR'2023 Editing models with task arithmetic. https://openreview.net/forum?id=6t0Kwf8-jrj
> [7]NeurIPS'2023 TIES-Merging: Resolving Interference When Merging Models. https://openreview.net/forum?id=xtaX3WyCj1
> [8]NeurIPS'2024 Twin-Merging: Dynamic Integration of Modular Expertise in Model Merging. https://openreview.net/forum?id=81YIt63TTn

---

> > ### Author Rebuttal · Reviewer_hasu · 2026-04-01
> >
> > The authors have addressed my concerns. I will raise the score from 3 to 4.

---

> > > ### Author Response · Authors · 2026-04-01
> > >
> > > Thank you for your time and for updating your score. We truly appreciate your careful evaluation and are glad that our rebuttal addressed your concerns.
> > >
> > > We will further improve the clarity of the paper in the final version.

---

### Decision · Program_Chairs · 2026-04-30

**Decision:**

Accept (regular)

**Comment:**

This submission proposes MetaMerging, a meta-learning framework that optimizes model merging coefficients for downstream adaptation rather than static multi-task performance. The paper received four positive reviews, and the authors' rebuttal effectively addressed reviewer concerns. The core idea is well-motivated, the method is lightweight and compatible with existing merging approaches, and experiments across vision and NLP benchmarks demonstrate consistent improvements. Given the positive scores from all reviewers, the recommendation is accept.